# Multi-block-Single-probe Variance Reduced Estimator for Coupled Compositional Optimization

**Wei Jiang[1], Gang Li[2], Yibo Wang[1], Lijun Zhang[1,*], Tianbao Yang[3,*]**
[1]National Key Laboratory for Novel Software Technology, Nanjing University, Nanjing, China
[2]Department of Computer Science, the University of Iowa, Iowa City, USA
[3]Department of Computer Science and Engineering, Texas A&M University, College Station, USA
jiangw@lamda.nju.edu.cn, gang-li@uiowa.edu.cn, wangyb@lamda.nju.edu.cn
zhanglj@lamda.nju.edu.cn, tianbao-yang@tamu.edu

## Abstract

Variance reduction techniques such as SPIDER/SARAH/STORM have been extensively studied to improve the convergence rates of stochastic non-convex optimization, which usually maintain and update a sequence of estimators for a single function across iterations. *What if we need to track multiple functional mappings across iterations but only with access to stochastic samples of $\mathcal{O}(1)$ functional mappings at each iteration?* There is an important application in solving an emerging family of coupled compositional optimization problems in the form of $\sum_{i=1}^{m} f_i(g_i(\mathbf{w}))$, where $g_i$ is accessible through a stochastic oracle. The key issue is to track and estimate a sequence of $\mathbf{g}(\mathbf{w}) = (g_1(\mathbf{w}), \ldots, g_m(\mathbf{w}))$ across iterations, where $\mathbf{g}(\mathbf{w})$ has $m$ blocks and it is only allowed to probe $\mathcal{O}(1)$ blocks to attain their stochastic values and Jacobians. To improve the complexity for solving these problems, we propose a novel stochastic method named Multi-block-Single-probe Variance Reduced (MSVR) estimator to track the sequence of $\mathbf{g}(\mathbf{w})$. It is inspired by STORM but introduces a customized error correction term to alleviate the noise not only in stochastic samples for the selected blocks but also in those blocks that are not sampled. With the help of the MSVR estimator, we develop several algorithms for solving the aforementioned compositional problems with improved complexities across a spectrum of settings with non-convex/convex/strongly convex/Polyak-Łojasiewicz (PL) objectives. Our results improve upon prior ones in several aspects, including the order of sample complexities and dependence on the strong convexity parameter. Empirical studies on multi-task deep AUC maximization demonstrate the better performance of using the new estimator.

## 1 Introduction

This paper is motivated by solving the following Finite-sum Coupled Compositional Optimization (FCCO) problem that has broad applications in machine learning [Wang and Yang, 2022]:

$$\min_{\mathbf{w} \in \mathbb{R}^d} F(\mathbf{w}) := \frac{1}{m} \sum_{i=1}^{m} f_i(g_i(\mathbf{w})), \tag{1}$$

where $f_i : \mathbb{R}^p \mapsto \mathbb{R}$ is a simple deterministic function. We assume that only noisy estimations of $g_i(\cdot)$ and its Jacobian $\nabla g_i(\cdot)$ can be accessed, denoted as $g_i(\cdot; \xi_i)$ and $\nabla g_i(\cdot; \xi_i)$, where $\xi_i$ represents the random sample(s) drawn from a stochastic oracle such that $\mathbb{E}\left[g_i(\cdot; \xi_i)\right] = g_i(\cdot)$ and

---

*Corresponding author

36th Conference on Neural Information Processing Systems (NeurIPS 2022).

$\mathbb{E}\left[\nabla g_i(\cdot; \xi_i)\right] = \nabla g_i(\cdot)$. A special case to be considered separately is when each $\xi_i$ has a finite support and is uniformly distributed. In this case, the problem can be represented as:

$$\min_{\mathbf{w} \in \mathbb{R}^d} F(\mathbf{w}) := \frac{1}{m} \sum_{i=1}^{m} f_i \left( \frac{1}{n} \sum_{j=1}^{n} g_i(\mathbf{w}; \xi_{ij}) \right). \tag{2}$$

These problems are different from classical stochastic compositional optimization (SCO) problems $\mathbb{E}_\zeta[f_\zeta(\mathbb{E}_\xi g(\mathbf{w}; \xi))]$ and its finite-sum variant $1/m \sum_{i=1}^{m} f_i(1/n \sum_{j=1}^{n} g(\mathbf{w}; \xi_j))$ [Wang et al., 2017], because the inner function is coupled with the outer index in FCCO.

A striking difference in solving FCCO problems is that we need to deal with multiple functional mappings of $g_i(\mathbf{w})$ for $i = 1, \ldots, m$. A challenge emerges when it is not possible to draw data samples for all blocks $i = 1, \ldots, m$ at each iteration due to some restrictions (e.g., limited memory and computational budget per-iteration). Wang and Yang [2022] studied this problem comprehensively and proposed an algorithm named as SOX. A key to their algorithmic design is to maintain and selectively update a sequence of estimators $\mathbf{u} = (\mathbf{u}^1, \ldots, \mathbf{u}^m)$ for tracking $\mathbf{g}(\mathbf{w}) = (g_1(\mathbf{w}), \ldots, g_m(\mathbf{w}))$ by exponential moving average, i.e.,

$$\mathbf{u}_t^i = \begin{cases} (1-\beta)\mathbf{u}_{t-1}^i + \beta g_i\left(\mathbf{w}_t; \xi_t^i\right) & i \in \mathcal{B}_1^t \\ \mathbf{u}_{t-1}^i & i \notin \mathcal{B}_1^t \end{cases}, \tag{3}$$

where $\xi_t^i$ and $\mathcal{B}_t^1 \subseteq \{1, \ldots, m\}$ denote a set of sampled blocks. With $\mathbf{u}$, the gradient estimator is computed by exponential moving average as well. As a result, they establish a sample complexity of $\mathcal{O}(m\epsilon^{-4})$ for non-convex objectives, $\mathcal{O}(m\epsilon^{-3})$ for convex objectives and $\mathcal{O}(m\mu^{-2}\epsilon^{-1})$ for $\mu$-strongly convex objectives. However, there are several caveats of these results: (i) the sample complexities (e.g., $\mathcal{O}(m\epsilon^{-4})$ for a non-convex objective) are no better than **probing all blocks** at each iteration, for which Ghadimi et al. [2020] have established an $\mathcal{O}(\epsilon^{-4})$ iteration complexity and an $\mathcal{O}(m\epsilon^{-4})$ sample complexity; (ii) when $m = |\mathcal{B}_t^1| = 1$, the problem reduces to a special case of classic SCO problems; however, the complexities are worse than the state-of-the-art (SOTA) sample complexities for non-convex, convex and strongly convex objectives, which are $\mathcal{O}(\epsilon^{-3})$, $\mathcal{O}(\epsilon^{-2})$ and $\mathcal{O}(\mu^{-1}\epsilon^{-1})$, respectively [Zhang and Xiao, 2019, Jiang et al., 2022]. A useful technique for achieving these complexities in prior works is by using variance reduction techniques, so a straightforward approach is to change the update of $\mathbf{u}_t^i$ by using a variance reduced estimator and do similarly for the gradient estimator. In particular, one can change the update for $\mathbf{u}_t^i$ according to STORM [Cutkosky and Orabona, 2019]:

$$\mathbf{u}_t^i = \begin{cases} (1-\beta)\mathbf{u}_{t-1}^i + \beta g_i\left(\mathbf{w}_t; \xi_t^i\right) + \underbrace{(1-\beta)(g_i\left(\mathbf{w}_t; \xi_t^i\right) - g_i\left(\mathbf{w}_{t-1}; \xi_t^i\right))}_{\text{error correction}} & i \in \mathcal{B}_1^t \\ \mathbf{u}_{t-1}^i & i \notin \mathcal{B}_1^t \end{cases}. \tag{4}$$

However, this simple change does not improve the complexities over that obtained by Wang and Yang [2022]. The reason is that the standard error correction term marked above in STORM only accounts for the randomness in $g_i(\mathbf{w}_t; \xi_t^i)$ but not in the randomness caused by sampling $i \in \mathcal{B}_1^t$. So, a major question remains:

> *How can we further improve the complexities for solving FCCO to match the SOTA results of SCO by using variance reduction techniques via probing only $\mathcal{O}(1)$ blocks at each iteration?*

To address this issue, we propose a novel variance reduction technique by selectively updating $\mathbf{u}_t^i$ for tracking $\mathbf{g}(\mathbf{w}_t)$, to which we refer as Multi-block-Single-probe variance-reduced (MSVR) estimator. It employs a similar update as STORM for selected $\mathbf{u}_t^i$ **but with a different customized error correction term** to deal with the randomness in both $g_i(\mathbf{w}_t; \xi_t^i)$ and that in $\mathcal{B}_1^t$. Based on MSVR, we develop several algorithms for FCCO problems with different ways to compute the gradients, and analyze the sample complexities across a spectrum of settings with non-convex/convex/strongly convex/PL objectives and finite/infinite support of $\xi_i$. We summarize our contributions and our results below:

- We develop a novel MSVR estimator for tracking a sequence of multiple blocks of functional mappings by only probing $\mathcal{O}(1)$ blocks via random samples at each iteration.

Table 1: Sample complexities needed to find an $\epsilon$-stationary point or $\epsilon$-optimal point. Here NC means non-convex, C means convex, SC indicates $\mu$-strongly convex, PL means the $\mu$-PL condition. $B_1$ denotes the outer batch size, i.e., $B_1 = |\mathcal{B}_t^1|$ and $B_2$ denotes the inner batch size. † assumes that $f$ is convex and monotone, and $g$ is convex but possibly not smooth. ∗ applies when inner function is in the form of the finite-sum. $\widetilde{\mathcal{O}}(\cdot)$ hides logarithmic factors. In all results, we assume $m \leq \mathcal{O}(\epsilon^{-1})$.

| Method | NC | C | SC/PL | $B_1, B_2$ |
|---|---|---|---|---|
| BSGD [Hu et al., 2020] | $\mathcal{O}\left(\epsilon^{-6}\right)$ | $\mathcal{O}\left(\epsilon^{-3}\right)$ | $\mathcal{O}\left(\mu^{-1}\epsilon^{-3}\right)$ (SC) | $\mathcal{O}(1), \mathcal{O}\left(\epsilon^{-2}\right)$(NC) $\mathcal{O}(1), \mathcal{O}\left(\epsilon^{-1}\right)$(C/SC) |
| BSpiderBoost [Hu et al., 2020] | $\mathcal{O}\left(\epsilon^{-5}\right)$ | - | - | $\mathcal{O}\left(\epsilon^{-1}\right), \mathcal{O}\left(\epsilon^{-2}\right)$ |
| SOX | $\mathcal{O}\left(m\epsilon^{-4}\right)$ | $\mathcal{O}\left(m\epsilon^{-3}\right)$ | $\mathcal{O}\left(m\mu^{-2}\epsilon^{-1}\right)$ | $\mathcal{O}(1), \mathcal{O}(1)$ |
| SOX ($\beta = 1$) [Wang and Yang, 2022] | - | $\mathcal{O}\left(mB_2\epsilon^{-2}\right)^{\dagger}$ | - | $\mathcal{O}(1), \mathcal{O}(1)$ |
| **MSVR-v1** | $\mathcal{O}\left(\max(B_1, B_2)\epsilon^{-4}\right)$ | $\mathcal{O}\left(\max(B_1, B_2)\epsilon^{-3}\right)$ | $\mathcal{O}\left(\max(B_1, B_2)\mu^{-2}\epsilon^{-1}\right)$ | $\mathcal{O}(1), \mathcal{O}(1)$ |
| **MSVR-v2** | $\mathcal{O}\left(m\sqrt{B_2}\epsilon^{-3}\right)$ | $\mathcal{O}\left(m\sqrt{B_2}\epsilon^{-2}\right)$ | $\mathcal{O}\left(m\sqrt{B_2}\mu^{-1}\epsilon^{-1}\right)$ | $\mathcal{O}(1), \mathcal{O}(1)$ |
| **MSVR-v3**∗ | $\mathcal{O}\left(m\sqrt{nB_2}\epsilon^{-2}\right)$ | $\widetilde{\mathcal{O}}\left(m\sqrt{nB_2}\epsilon^{-1}\right)$ | $\widetilde{\mathcal{O}}\left(m\sqrt{nB_2}\mu^{-1}\right)$ | $\mathcal{O}(1), \mathcal{O}(1)$ |

- By applying the MSVR estimator, we develop three algorithms for FCCO by using different methods for computing the gradients, and establish improved complexities for non-convex, convex, strongly convex, and PL objectives. A comparison between our algorithms and existing methods is shown in Table 1, where we also exhibit the dependence on $B_2$, which is the size of the inner batch for estimating each $g_i(\mathbf{w})$.

- The complexity of our first method (i.e., MSVR-v1) enjoys the same order on $\epsilon$ as SOX, but does not depend on $m$; MSVR-v2 improves the dependence on $\epsilon$, and its complexities match the SOTA results for SCO when $m = 1$; our MSVR-v3 further reduces the dependence on $\epsilon$ for the finite support of $\xi$, and also attains the SOTA complexities when $m = 1$.

- We conduct experiments on multi-task deep AUC maximization to verify the theory and demonstrate the advantage of the proposed algorithms.

## 2  Related work

This section briefly reviews related work on variance-reduced methods and stochastic compositional optimization (SCO) problems.

Variance-reduction (VR) techniques for improving the convergence of stochastic optimization originate from Roux et al. [2012] for solving convex finite-sum empirical risk minimization (ERM) problems. Since then, different VR techniques have been proposed for convex finite-sum ERM, e.g., SVRG [Johnson and Zhang, 2013, Zhang et al., 2013] and SAGA [Defazio et al., 2014]. These works have improved the complexity for solving smooth and strongly convex problems to a logarithmic complexity. For non-convex ERM problems, Fang et al. [2018] invents the SPIDER estimator similar to its predecessor SARAH [Nguyen et al., 2017], and improve the complexity of standard SGD from $O(\epsilon^{-4})$ to $O(\epsilon^{-3})$ and $O(\sqrt{n}\epsilon^{-2})$ in stochastic and finite-sum settings, respectively, where $n$ is the number of components in the finite-sum. Algorithmic improvements have been made to SPIDER by using a constant step size in SpiderBoost [Wang et al., 2018] and using a constant batch size in STORM [Cutkosky and Orabona, 2019].

Several classes of SCO have been studied. The first class is the two-level SCO whose objective is given by $\mathbb{E}_\xi[f_\xi(\mathbb{E}_\omega[g_\omega(\mathbf{w})])]$, where $\xi$ and $\omega$ are random variables. While the study of two-level compositional functions dates back to the 70s, the most recent comprehensive study was initiated by Wang et al. [2017]. They proposed a two time-scale classic algorithm named SCGD and establish its asymptotic guarantee and non-asymptotic convergence rates. Following this work, many studies have been devoted to improving the convergence rates or algorithmic design of two-level SCO [Wang et al., 2016, Ghadimi et al., 2020, Zhang and Lan, 2021]. In particular, recent works have used variance-reduction techniques based on SPIDER/SARAH/STORM to estimate the inner values and the gradients [Liu et al., 2018, Yuan et al., 2019a, Zhang and Xiao, 2019, Chen et al., 2021, Qi et al., 2021a]. Similar efforts have been extended to the second class of SCO, i.e., multi-level SCO with an

objective $\mathbb{E}_{\xi_1}[f^1_{\xi_1}(\mathbb{E}_{\xi_2}[f^2_{\xi_2}(\ldots(\mathbb{E}_{\xi_K}[f^K_{\xi_K}(\mathbf{w})])\ldots)])]$ [Yang et al., 2019]. Recent studies have been focused on further improving the sample complexity and reducing the dependence on the number of levels $K$ [Balasubramanian et al., 2021, Chen et al., 2021, Zhang and Lan, 2021, Zhang and Xiao, 2021, Jiang et al., 2022]. These works also employed variance reduction techniques to design their own methods. However, directly applying these algorithms of two-level and multi-level SCO to FCCO requires probing all $m$ blocks in $\mathbf{g}(\mathbf{w})$, which is prohibitive in many applications.

The third class of SCO is the Conditional Stochastic Optimization (CSO) whose objective is in the form of $\mathbb{E}_\xi[f_\xi(\mathbb{E}_{\omega|\xi}g_\omega(\mathbf{w};\xi))]$ [Hu et al., 2020], where $\omega|\xi$ means that the distribution of $\omega$ might depend on $\xi$. The FCCO problem can be considered as a special case of CSO. The key difference from the first class of SCO discussed above is that the inner function $g$ depends on the random variable $\xi$ of the outer level. For CSO, Hu et al. [2020] proposed two algorithms with and without using the variance-reduction technique (SpiderBoost) named BSGD and BSpiderboost, and established complexities for non-convex, convex and strongly convex functions, which are shown in Table 1. However, their algorithms require a large batch size for estimating the inner functions.

Recently, a novel class (the fourth class) of SCO was studied, which is referred to as the finite-sum coupled compositional optimization (FCCO) [Wang and Yang, 2022]. The finite-sum structure makes it possible to develop more practical algorithms without relying on huge batch size per-iteration. It was first studied by Qi et al. [2021b] for maximizing the point-estimator of the area under the precision-recall curve. Recently, it was comprehensively investigated by Wang and Yang [2022] and more applications of FCCO have been demonstrated in machine learning. Nevertheless, their algorithm—SOX does not use variance reduction techniques and hence suffers from the limitations discussed in the previous section.

## 3 Proposed Algorithms and Convergence

First, we introduce the notations and assumptions used in this paper. Then we describe the MSVR estimator in detail and develop algorithms based on the proposed estimator.

### 3.1 Notations and Assumptions

Let $[m] = \{1, \ldots, m\}$. The definition of sample complexity is given below, which is widely used to measure the efficiency of stochastic algorithms.

**Definition 1** *The sample complexity is the number of samples needed to find a point satisfying* $\mathbb{E}\left[\|\nabla F(\mathbf{w})\|\right] \leq \epsilon$ *($\epsilon$-stationary) or* $\mathbb{E}\left[F(\mathbf{w}) - \inf_{\mathbf{w}} F(\mathbf{w})\right] \leq \epsilon$ *($\epsilon$-optimal).*

Next, we make following assumptions throughout the paper, which are commonly used in the studies of SCO [Wang et al., 2016, 2017, Yuan et al., 2019a, Zhang and Xiao, 2019, 2021].

**Assumption 1** *(Smoothness and Lipschitz continuity) We assume that each $f_i$ is $L_f$-smooth and $C_f$-Lipchitz continuous, each $g_i$ is $L_g$-smooth and $C_g$-Lipchitz continuous.*

**Remark:** This implies $F(\mathbf{w})$ is $C_F$-Lipschitz continuous and $L_F$-smooth, where $C_F = C_f C_g$, $L_F = C_f^2 L_g + C_g^2 L_f$ [Zhang and Xiao, 2021].

**Assumption 2** *(Bounded variance)*

$$\mathbb{E}\left[g_i(\mathbf{x};\xi_t^i)\right] = g_i(\mathbf{x}); \qquad\qquad \mathbb{E}\left[\nabla g_i(\mathbf{x};\xi_t^i)\right] = \nabla g_i(\mathbf{x});$$

$$\mathbb{E}\left[\left\|g_i\left(\mathbf{x};\xi_t^i\right) - g_i(\mathbf{x})\right\|^2\right] \leq \sigma^2/B_2; \qquad \mathbb{E}\left[\left\|\nabla g_i\left(\mathbf{x};\xi_t^i\right) - \nabla g_i(\mathbf{x})\right\|^2\right] \leq \sigma^2/B_2;$$

*where the random variable $\xi_t^i$ denotes a batch of samples with batch size $B_2 \geq 1$.*

**Assumption 3** *(Average Lipchitz continuity of $g_i$ and its Jacobian)*

$$\mathbb{E}\left[\left\|g_i\left(\mathbf{x};\xi_t^i\right) - g_i\left(\mathbf{y};\xi_t^i\right)\right\|^2\right] \leq C_g^2\|\mathbf{x}-\mathbf{y}\|^2;$$

$$\mathbb{E}\left[\left\|\nabla g_i\left(\mathbf{x};\xi_t^i\right) - \nabla g_i\left(\mathbf{y};\xi_t^i\right)\right\|^2\right] \leq L_g^2\|\mathbf{x}-\mathbf{y}\|^2.$$

**Remark:** Although this assumption seems strong at the first sight, it is quite standard and widely used in the recent compositional optimization literature [Yuan et al., 2019a, Zhang and Xiao, 2019, 2021, Jiang et al., 2022].

**Assumption 4** $F_* = \inf_{\mathbf{w}} F(\mathbf{w}) \geq -\infty$ *and* $F(\mathbf{w}_1) - F_* \leq \Delta_F$ *for the initial solution* $\mathbf{w}_1$.

## 3.2 Multi-block-Single-probe Variance Reduced (MSVR) Estimator

Assume that we have a budget to probe only $B_1$ out of $m$ mappings in $\mathbf{g}(\mathbf{w})$. To this end, at the $t$-th iteration we sample a set of blocks $\mathcal{B}_1^t \subseteq [m]$, where $|\mathcal{B}_1^t| = B_1$, and probe the corresponding $g_i(\mathbf{w})$ by accessing the noisy estimates $g_i(\mathbf{w}_t; \xi_t^i)$ for $i \in \mathcal{B}_1^t$. Then, we just update the corresponding block in our estimator $\mathbf{u}_t$. Specifically, we update $\mathbf{u}_t^i$ for $i \in \mathcal{B}_1^t$ in a new way and keep other blocks unchanged. The whole estimator is shown below:

$$\mathbf{u}_t^i = \begin{cases} \underbrace{(1 - \beta_t)\mathbf{u}_{t-1}^i + \beta_t g_i(\mathbf{w}_t; \xi_t^i) + \boxed{\gamma_t}\left(g_i(\mathbf{w}_t; \xi_t^i) - g_i(\mathbf{w}_{t-1}; \xi_t^i)\right)}_{\bar{\mathbf{u}}_i^t} & i \in \mathcal{B}_1^t \\ \mathbf{u}_{t-1}^i & i \notin \mathcal{B}_1^t \end{cases} . \quad (5)$$

The first line of our estimator is inspired by STORM [Cutkosky and Orabona, 2019]. The difference is that the STORM estimator sets $\gamma_t = (1 - \beta_t)$, while for MSVR, $\gamma_t$ is set as $\frac{m - B_1}{B_1(1 - \beta_t)} + (1 - \beta_t)$ according to our analysis. We name equation (5) as Multi-block-Single-probe Variance Reduced (MSVR) estimator. By multi-block, we mean the estimator can track multiple functional mappings $(g_1, g_2, \cdots, g_m)$, simultaneously; by single-probe, we indicate the number of sampled blocks $B_1$ for probing can be as small as one. It is notable that when $B_1 = m$, i.e., all blocks are probed at each iteration, $\gamma_t = 1 - \beta_t$ and MSVR reduces to STORM applied to $\mathbf{g}(\mathbf{w})$. The additional factor in $\gamma_t$, i.e., $\gamma_t^0 = \frac{m - B_1}{B_1(1 - \beta_t)}$ is to account for the randomness in the sampled blocks and noise in those blocks that are not updated. To briefly understand the additional factor $\gamma_t^0$, we consider bounding $\|\mathbf{u}_t - \mathbf{g}(\mathbf{w}_t)\|^2 = \sum_{i=1}^m \|\mathbf{u}_t^i - g_i(\mathbf{w}_t)\|^2$. Let us focus on a fixed $i \in [m]$. Then we have

$$\mathbb{E}\left[\|\mathbf{u}_t^i - g_i(\mathbf{w}_t)\|^2\right] = \frac{B_1}{m} \underbrace{\mathbb{E}\left[\|\bar{\mathbf{u}}_t^i - g_i(\mathbf{w}_t)\|^2\right]}_{A_1} + (1 - \frac{B_1}{m}) \underbrace{\mathbb{E}\left[\|\mathbf{u}_{t-1}^i - g_i(\mathbf{w}_t)\|^2\right]}_{A_2}.$$

Note that the first term $A_1$ in the R.H.S. can be bounded similarly as STORM by building recurrence with $\|\mathbf{u}_{t-1}^i - g_i(\mathbf{w}_{t-1})\|^2$. However, there exists the second term due to the randomness of $\mathcal{B}_1^t$, which can be decomposed as

$$\|\mathbf{u}_{t-1}^i - g_i(\mathbf{w}_{t-1}) + g_i(\mathbf{w}_{t-1}) - g_i(\mathbf{w}_t)\|^2 = \underbrace{\|\mathbf{u}_{t-1}^i - g_i(\mathbf{w}_{t-1})\|^2}_{A_{21}} + \underbrace{\|g_i(\mathbf{w}_{t-1}) - g_i(\mathbf{w}_t)\|^2}_{A_{22}}$$
$$+ \underbrace{2(\mathbf{u}_{t-1}^i - g_i(\mathbf{w}_{t-1}))^\top (g_i(\mathbf{w}_{t-1}) - g_i(\mathbf{w}_t))}_{A_{23}}.$$

The first two terms in R.H.S. ($A_{21}$ and $A_{22}$) can be easily handled. The difficulty comes from the third term, which cannot be simply bounded by using Young's inequality. If doing so, it will end up with a non-diminishing error of $\mathbf{u}_t^i$. To combat this difficulty, we use the additional factor brought by $\gamma_t^0(g_i(\mathbf{w}_t; \xi_t^i) - g_i(\mathbf{w}_{t-1}; \xi_t^i))$ in $A_1$ to cancel $A_{23}$. This is more clear by the following decomposition of $A_1$.

$$A_1 = \mathbb{E}[\|\underbrace{(1 - \beta_t)(\mathbf{u}_{t-1}^i - g_i(\mathbf{w}_{t-1}))}_{A_{11}} + \underbrace{\gamma_t^0(g_i(\mathbf{w}_t) - g_i(\mathbf{w}_{t-1}))}_{A_{12}}$$
$$+ \underbrace{\beta_t(g_i(\mathbf{w}_t; \xi_t^i) - g_i(\mathbf{w}_t))}_{A_{13}} + \underbrace{\gamma_t(g_i(\mathbf{w}_t; \xi_t^i) - g_i(\mathbf{w}_{t-1}; \xi_t^i) - g_i(\mathbf{w}_t) + g_i(\mathbf{w}_{t-1}))}_{A_{14}}\|^2]$$
$$\leq \mathbb{E}[\|A_{11} + A_{12}\|^2] + \mathbb{E}\left[\|A_{13} + A_{14}\|^2\right].$$

In light of the above decomposition, we can bound $\mathbb{E}[\|A_{11} + A_{12}\|^2] \leq \mathbb{E}[\|A_{11}\|^2 + \|A_{12}\|^2 + 2A_{11}^\top A_{12}]$ and $\mathbb{E}[\|A_{13} + A_{14}\|^2] \leq 2\mathbb{E}[\|A_{13}\|^2] + 2\mathbb{E}[\|A_{14}\|^2]$. The resulting term $\mathbb{E}[2A_{11}^\top A_{12}]$ has a negative sign as $A_{23}$. Hence, by carefully choosing $\gamma_t^0$, we can cancel both terms. The remaining terms can be organized similarly as in the analysis for STORM. We give a technical lemma for building the recurrence of MSVR's error below. All the proofs are deferred to the supplementary material due to space limitations.

**Lemma 1** *By setting $\gamma_t = \frac{m - B_1}{B_1(1 - \beta_t)} + (1 - \beta_t)$, for $\beta_t \leq \frac{1}{2}$, we have:*

$$\mathbb{E}\left[\|\mathbf{u}_t - g(\mathbf{w}_t)\|^2\right] \leq \left(1 - \frac{B_1\beta_t}{m}\right)\mathbb{E}\left[\|\mathbf{u}_{t-1} - g(\mathbf{w}_{t-1})\|^2\right] + \frac{2B_1\beta_t^2\sigma^2}{B_2}$$
$$+ \frac{8m^2 C_g^2}{B_1}\mathbb{E}\left[\|\mathbf{w}_t - \mathbf{w}_{t-1}\|^2\right].$$

---
**Algorithm 1** MSVR-v1 and MSVR-v2 method
---
1: **Input:** time step $T$, parameters $\alpha_t$, $\beta_t$, $\gamma_t$, learning rate $\eta_t$ and initial points $(\mathbf{w}_1, \mathbf{u}_1, \mathbf{z}_1)$.
2: **for** time step $t = 1$ **to** $T$ **do**
3:      Sample a subset $\mathcal{B}_1^t$ from $\{1, 2, \cdots, m\}$
4:      Compute estimator $\mathbf{u}_t$ according to equation (5) or (6)     $\diamond$ Use MSVR or MSVR-SP update
5:      (v1) Compute estimator $\mathbf{z}_t$ according to equation (7)         $\diamond$ Use moving average update
6:      (v2) Compute estimator $\mathbf{z}_t$ according to equation (8)              $\diamond$ Use STORM update
7:      $\mathbf{w}_{t+1} = \mathbf{w}_t - \eta_t \mathbf{z}_t$
8: **end for**
9: Choose $\tau$ uniformly at random from $\{1, \ldots, T\}$
10: Return $\mathbf{w}_\tau$
---

**Remark:** The above recursion is similar to that of STORM for tracking a sequence of a single-block functional mapping. Since the last term $\|\mathbf{w}_t - \mathbf{w}_{t-1}\|^2$ can be offset in the future analysis, intuitively the estimation error $\|\mathbf{u}_t - g(\mathbf{w}_t)\|^2$ would reduce after each iteration.

**Single Point Version.** A limitation of the MSVR estimator is that it needs to probe selected blocks at two different points, i.e., $g_i(\mathbf{w}_t; \xi_t^i)$ and $g_i(\mathbf{w}_{t-1}; \xi_t^i)$. With a more careful analysis, we can probe a selected block at a single point similar to that used by Balasubramanian et al. [2021] and Chen et al. [2021]. Specifically, we replace $g_i(\mathbf{w}_t; \xi_t^i) - g_i(\mathbf{w}_{t-1}; \xi_t^i)$ with $\nabla g_i(\mathbf{w}_t; \xi_t^i)^\top (\mathbf{w}_t - \mathbf{w}_{t-1})$. As a result, we propose a single-point version of MSVR (named as MSVR-SP) estimator below:

$$\mathbf{u}_t^i = \begin{cases} (1 - \beta_t)\mathbf{u}_{t-1}^i + \beta_t g_i(\mathbf{w}_t; \xi_t^i) + \gamma_t \nabla g_i(\mathbf{w}_t; \xi_t^i)^\top (\mathbf{w}_t - \mathbf{w}_{t-1}) & i \in \mathcal{B}_1^t \\ \mathbf{u}_{t-1}^i & i \notin \mathcal{B}_1^t \end{cases}. \quad (6)$$

The MSVR-SP estimator enjoys the similar recurrence for the estimation error.

**Lemma 2** *Set* $\gamma_t = \frac{m - B_1}{B_1(1 - \beta_t)} + (1 - \beta_t)$. *If* $\|\mathbf{w}_{t+1} - \mathbf{w}_t\|^2 \leq \eta_t^2 C_F^2$ *and* $\eta_t \leq \sqrt{\beta_t}$, *we have:*

$$\mathbb{E}\left[\|\mathbf{u}_t - g(\mathbf{w}_t)\|^2\right] \leq \left(1 - \frac{B_1 \beta_t}{m}\right) \mathbb{E}\left[\|\mathbf{u}_{t-1} - g(\mathbf{w}_{t-1})\|^2\right] + \frac{2B_1 \beta_t^2 \sigma^2}{B_2}$$

$$+ \left(4L_g^2 C_F^2 + 9C_g^2 + \frac{8\sigma^2}{B_2}\right) \frac{m^2}{B_1} \mathbb{E}\left[\|\mathbf{w}_t - \mathbf{w}_{t-1}\|^2\right].$$

**Remark:** If there is a constraint on the range of $g_i$, we can add a projection to the update of $\mathbf{u}_t^i$ such that it always resides in the range, which will not affect the analysis of Lemma 1 and Lemma 2.

### 3.3 Leveraging the MSVR Estimator for solving the FCCO Problem

Now, we are ready to present our proposed algorithms for solving problem (1). The first two algorithms (named MSVR-v1 and MSVR-v2) are presented in Algorithm 1. These two methods differ in how to estimate the gradient.

Let us first consider MSVR-v1. At each time step $t$, we first use the proposed MSVR or MSVR-SP estimator $\mathbf{u}_t$ to estimate the inner function value. Then, following the previous literature [Wang et al., 2021, Wang and Yang, 2022], we use the moving average estimator $\mathbf{z}_t$ to estimate the gradient as:

$$\mathbf{z}_t = \Pi_{C_F}\left[(1 - \alpha_t)\mathbf{z}_{t-1} + \frac{\alpha_t}{B_1} \sum_{i \in \mathcal{B}_1^t} \nabla f_i(\mathbf{u}_{t-1}^i)\nabla g_i(\mathbf{w}_t; \xi_t^i)\right], \quad (7)$$

where $\Pi_{C_F}$ denotes the projection onto the ball with radius $C_F$. This projection **is optional** for using MSVR, but is required for using MSVR-SP to ensure $\|\mathbf{w}_{t+1} - \mathbf{w}_t\|^2 \leq \eta_t^2 C_F^2$ as used in Lemma 2. Since the true gradient $\nabla F$ is also in this ball, i.e., $\|\nabla F\| \leq C_F$, the projection will not affect the future analysis. Also note that when computing the estimator $\mathbf{z}_t$, we use $\nabla f_i(\mathbf{u}_{t-1}^i)$ instead of $\nabla f_i(\mathbf{u}_t^i)$ to avoid the dependence on the random variable $\xi_t^i$, which may lead to dependent issues otherwise. Finally, we use the estimated gradient $\mathbf{z}_t$ to update the parameter $\mathbf{w}_{t+1}$. Now, we provide the theoretical guarantee for the MSVR-v1 method.

**Theorem 1** *Our MSVR-v1 algorithm with* $\alpha_{t+1} = \mathcal{O}\left(\eta_t\right)$, $\beta_{t+1} = \mathcal{O}(\frac{m^2\eta_t^2}{B_1^2})$, $a = \mathcal{O}(\frac{mB_2}{B_1})$ *and* $\eta_t = \min\left\{\left(\frac{B_1\sqrt{B_2}}{m}\right)^{2/3}(a+t)^{-1/3}, \sqrt{\min\{B_1,B_2\}}(a+t)^{-1/2}\right\}$, *can find an $\epsilon$-stationary point in* $\mathcal{O}\left(\max\left\{\frac{m\epsilon^{-3}}{B_1\sqrt{B_2}}, \frac{\epsilon^{-4}}{\min\{B_1,B_2\}}\right\}\right)$ *iterations.*

**Remark:** This complexity is strictly better than previous SOTA method SOX, which enjoys an iteration complexity of $\mathcal{O}\left(\max\left\{\frac{m\epsilon^{-4}}{B_1 B_2}, \frac{\epsilon^{-4}}{\min\{B_1,B_2\}}, \frac{m\epsilon^{-2}}{B_1}\right\}\right)$. The sample complexity can be obtained by multiplying the iteration complexity with $B_1 B_2$. We can see that larger $B_1$ or $B_2$ yields a smaller iteration complexity, which means that from the computational perspective, if samples can be processed in parallel (e.g., in GPU), there is a benefit of using large $B_1$ and/or $B_2$. However, from the sample complexity perspective, using $B_1 = B_2 = 1$ is the best. The same discussion holds for other theorems below.

However, the complexity of MSVR-v1 is still on the order of $\mathcal{O}(\epsilon^{-4})$. Due to the biased nature of the estimated gradient, using the moving average update is not enough for achieving the SOTA complexity of $\mathcal{O}(\epsilon^{-3})$. So, we use the technique of STORM [Cutkosky and Orabona, 2019] to update $\mathbf{z}_t$ as follows:

$$
\mathbf{z}_t = \Pi_{C_F}\left[(1-\alpha_t)\mathbf{z}_{t-1} + \alpha\frac{1}{B_1}\sum_{i\in\mathcal{B}_1^t}\nabla f_i(\mathbf{u}_{t-1}^i)\nabla g_i(\mathbf{w}_t;\xi_t^i)\right.
$$
$$
\left. + (1-\alpha_t)\frac{1}{B_1}\sum_{i\in\mathcal{B}_1^t}\left(\nabla f_i(\mathbf{u}_{t-1}^i)\nabla g_i(\mathbf{w}_t;\xi_t^i) - \nabla f_i(\mathbf{u}_{t-2}^i)\nabla g_i(\mathbf{w}_{t-1};\xi_t^i)\right)\right], \tag{8}
$$

where the projection operation is needed if using the MSVR estimator. Now, we prove this new method (i.e., MSVR-v2) can obtain the optimal complexity of $\mathcal{O}(\epsilon^{-3})$.

**Theorem 2** *Our MSVR-v2 algorithm with* $\alpha_{t+1} = \mathcal{O}(\frac{m\eta_t^2}{B_1})$, $\beta_{t+1} = \mathcal{O}\left(\frac{m^2\eta_t^2}{B_1^2}\right)$, $a = O(\frac{mB_2}{B_1})$ *and* $\eta_t = \mathcal{O}\left((\frac{B_1\sqrt{B_2}}{m})^{2/3}(a+t)^{-1/3}\right)$, *can find an $\epsilon$-stationary point in* $\mathcal{O}\left(\frac{m\epsilon^{-3}}{B_1\sqrt{B_2}}\right)$ *iterations.*

**Remark:** When $m = 1$ and $f$ is the identity function, problem (1) reduces to the standard stochastic non-convex optimization, whose lower bound is $\Omega\left(\epsilon^{-3}\right)$ [Arjevani et al., 2019], indicating our MSVR-v2 is optimal.

Next, we show that the complexity can be further improved when the objective function is convex or strongly convex. We note that Polyak-Łojasiewicz (PL) [Karimi et al., 2016] objectives are more general than strongly convex functions, since $\mu$-strong convexity implies the $\mu$-PL condition. So, we will consider the PL condition and introduce its definition below.

**Definition 2** $F(\mathbf{w})$ *satisfies the $\mu$-PL condition if there exists $\mu > 0$ such that:*

$$
2\mu\left(F(\mathbf{w}) - F_*\right) \leq \|\nabla F(\mathbf{w})\|^2.
$$

Then, we derive improved rates for convex or PL objectives by using the stage-wise design given in Algorithm 3 in the supplement.

**Theorem 3** *If the objective function satisfies the convexity or $\mu$-PL condition, MSVR-v1 derives a sample complexity of $\mathcal{O}(\max(B_1, B_2)\epsilon^{-3})$ or $\mathcal{O}(\max(B_1, B_2)\mu^{-2}\epsilon^{-1})$, separately. For MSVR-v2, the complexity can be further improved to $\mathcal{O}\left(m\sqrt{B_2}\epsilon^{-2}\right)$ or $\mathcal{O}\left(m\sqrt{B_2}\mu^{-1}\epsilon^{-1}\right)$.*

**Remark:** The complexities for MSVR-v2 are optimal, since they match the $\Omega\left(\epsilon^{-2}\right)$ and $\Omega\left(\mu^{-1}\epsilon^{-1}\right)$ lower bound for stochastic convex and strongly convex optimization [Agarwal et al., 2012].

**Remark:** The algorithms proposed in this paper can also use adaptive (Adam-style) learning rates and obtain the same complexity using the techniques proposed by Guo et al. [2021]. The details are provided in the supplementary.

---

**Algorithm 2** MSVR-v3 method

---
1: **Input:** time step $T$, parameters $\alpha$, $\beta$,$\gamma$, $I$, learning rate $\eta$ and initial points $(\mathbf{w}_1, \mathbf{u}_1, \mathbf{z}_1)$.
2: **for** time step $t = 1$ **to** $T$ **do**
3:     **if** $t \mod I == 0$ **then**
4:         Set $\tau = t$
5:         Compute and save $g_i(\mathbf{w}_\tau), \nabla f_i(\mathbf{u}_{\tau-1}^i)$ for every $i$ and $\frac{1}{m}\sum_{i=1}^m \nabla f_i(\mathbf{u}_{\tau-1}^i)\nabla g_i(\mathbf{w}_\tau)$
6:     **end if**
7:     Sample a subset $\mathcal{B}_1^t$ from $\{1, 2, \cdots, m\}$
8:     Compute function value estimator $\mathbf{u}_t$ according to equation (9)
9:     Compute gradient estimator $\mathbf{z}_t$ according to equation (10)
10:    $\mathbf{w}_{t+1} = \mathbf{w}_t - \eta\mathbf{z}_t$
11: **end for**
12: Choose $\tau$ uniformly at random from $\{1, \ldots, T\}$
13: Return $\mathbf{w}_\tau$

---

## 4 An Improved Rate for the Finite-sum Case

In this section, we consider the case that inner function $g_i$ is in the form of the finite-sum, i.e., $g_i(\mathbf{w}) = \frac{1}{n}\sum_{j=1}^n g_i(\mathbf{w}; \xi_{ij})$, so that we can compute the exact value of $g_i(\mathbf{w})$ in some iterations. We first modify our MSVR estimator to utilize the finite-sum structure. Inspired by SVRG [Johnson and Zhang, 2013, Zhang et al., 2013], we compute a full version of the inner function value for every $I$ iterations at $\mathbf{w}_\tau$, i.e., $g_i(\mathbf{w}_\tau) = \frac{1}{n}\sum_{j=1}^n g_i(\mathbf{w}_\tau; \xi_{ij})$ for $i = 1, \cdots, m$, where $\tau \mod I = 0$. Then, in each step, we use

$$\widehat{g}_i(\mathbf{w}_t; \xi_t^i) = g_i(\mathbf{w}_t; \xi_t^i) - g_i(\mathbf{w}_\tau; \xi_t^i) + g_i(\mathbf{w}_\tau)$$

to replace $g_i(\mathbf{w}_t; \xi_t^i)$ in the origin estimator. In this way, our MSVR estimator is changed to:

$$\mathbf{u}_t^i = \begin{cases} (1-\beta)\mathbf{u}_{t-1}^i + \beta\widehat{g}_i\left(\mathbf{w}_t; \xi_t^i\right) + \gamma\left(g_i\left(\mathbf{w}_t; \xi_t^i\right) - g_i\left(\mathbf{w}_{t-1}; \xi_t^i\right)\right) & i \in \mathcal{B}_1^t \\ \mathbf{u}_{t-1}^i & i \notin \mathcal{B}_1^t \end{cases}. \quad (9)$$

For this estimator, we have the following guarantee.

**Lemma 3** *If $\beta \leq \frac{1}{2}$ and $\beta I \leq \frac{m}{B_1}$, by setting $\gamma = \frac{m-B_1}{B_1(1-\beta)} + (1-\beta)$, we have:*

$$\mathbb{E}\left[\|\mathbf{u}_{t+1} - g\left(\mathbf{w}_{t+1}\right)\|^2\right] \leq \left(1 - \frac{B_1\beta}{m}\right)\mathbb{E}\left[\|\mathbf{u}_t - g\left(\mathbf{w}_t\right)\|^2\right] + \frac{10m^2 C_g^2}{B_1}\mathbb{E}\left[\|\mathbf{w}_{t+1} - \mathbf{w}_t\|^2\right].$$

**Remark:** Compared with Lemma 1, we remove the $\frac{2B_1\beta^2\sigma^2}{B_2}$ term, which is the key to reduce the complexity since we can now use a larger parameter $\beta$.

To attain the optimal complexity, we modify the gradient estimator $\mathbf{z}_t$ in a similar way:

$$\begin{aligned}\mathbf{z}_t = {} & (1-\alpha)\mathbf{z}_{t-1} + \alpha\mathbf{h}_t \\ & + (1-\alpha)\frac{1}{B_1}\sum_{i\in\mathcal{B}_1^t}\left(\nabla f_i(\mathbf{u}_{t-1}^i)\nabla g_i(\mathbf{w}_t; \xi_t^i) - \nabla f_i(\mathbf{u}_{t-2}^i)\nabla g_i(\mathbf{w}_{t-1}; \xi_t^i)\right),\end{aligned} \quad (10)$$

where $\mathbf{h}_t$ involves both the full gradient and the stochastic gradient (we also need to save each $\nabla f_i(\mathbf{u}_{\tau-1})$ and calculate the full version of $\frac{1}{m}\sum_{i=1}^m \nabla f_i(\mathbf{u}_{\tau-1}^i)\nabla g_i(\mathbf{w}_\tau)$ at those steps $\tau$) :

$$\mathbf{h}_t = \frac{1}{B_1}\sum_{i\in\mathcal{B}_1^t}\left(\nabla f_i(\mathbf{u}_{t-1}^i)\nabla g_i(\mathbf{w}_t; \xi_t^i) - \nabla f_i(\mathbf{u}_{\tau-1}^i)\nabla g_i(\mathbf{w}_\tau; \xi_t^i)\right) + \frac{1}{m}\sum_{i=1}^m \nabla f_i(\mathbf{u}_{\tau-1}^i)\nabla g_i(\mathbf{w}_\tau).$$

The whole method is summarized in Algorithm 2 (named as MSVR-v3). Next, we show that MSVR-v3 is equipped with an optimal complexity of $\mathcal{O}(\sqrt{n}\epsilon^{-2})$.

**Theorem 4** *Our MSVR-v3 with $I = \frac{mn}{B_1 B_2}$, $\alpha = \mathcal{O}\left(\frac{B_1 B_2}{mn}\right)$, $\beta = \mathcal{O}\left(\frac{B_2}{n}\right)$ and $\eta = \mathcal{O}\left(\frac{B_1\sqrt{B_2}}{m\sqrt{n}}\right)$, can obtain an $\epsilon$-stationary point in $T = \mathcal{O}\left(\frac{m\sqrt{n}\epsilon^{-2}}{B_1\sqrt{B_2}}\right)$ iterations.*

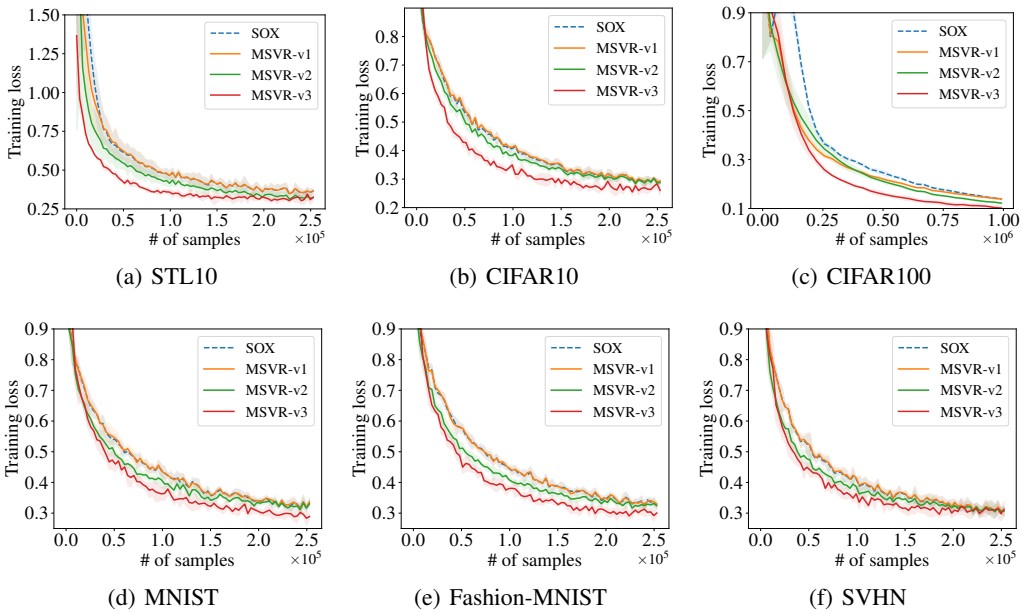

Figure 1: Results for Multi-task AUC Optimization.

**Remark:** When $m = 1$ and $f$ is the identity function, problem (2) reduces to the stochastic finite-sum optimization, whose optimal complexity is $\mathcal{O}\left(\sqrt{n}\epsilon^{-2}\right)$ [Fang et al., 2018, Li et al., 2021], indicating our complexity is optimal in terms of $\epsilon$ and $n$.

Similarly, a better complexity can be obtained under the convexity or PL condition.

**Theorem 5** *If the objective function satisfies the convexity or $\mu$-PL condition, the sample complexity can be improved to $\mathcal{O}\left(\frac{m\sqrt{n}\epsilon^{-1}}{B_1\sqrt{B_2}}\log\frac{1}{\epsilon}\right)$ or $\mathcal{O}\left(\frac{m\sqrt{n}\mu^{-1}}{B_1\sqrt{B_2}}\log\frac{1}{\epsilon}\right)$, respectively.*

**Remark:** It is notable that we achieve a linear convergence rate $\mathcal{O}\left(\log\frac{1}{\epsilon}\right)$ under the PL condition, matching the current result in the single-level finite-sum problem [Li et al., 2021]

## 5 Experiments

In this section, we conduct experiments on the multi-task deep AUC maximization to evaluate the proposed methods and we will consider more applications in the long version of the paper. For binary classification (label $y = 1$ or $y = -1$), AUC maximization can be formulated as minimizing the following composite loss [Zhu et al., 2022]:

$$\min_{\mathbf{w},a,b} \mathrm{E}_{\mathbf{x}|y=1}\left[(h_{\mathbf{w}}(\mathbf{x}) - a)^2\right] + \mathrm{E}_{\mathbf{x}'|y'=-1}\left[(h_{\mathbf{w}}(\mathbf{x}') - b)^2\right] + \ell(a(\mathbf{w}) - b(\mathbf{w})),$$

where $a(\mathbf{w}) = \mathrm{E}\left[h_{\mathbf{w}}(\mathbf{x}) \mid y = 1\right]$, $b(\mathbf{w}) = \mathrm{E}\left[h_{\mathbf{w}}(\mathbf{x}) \mid y = -1\right]$ and $\ell(\cdot)$ is a surrogate function. The above objective recovers the pairwise square loss and the min-max margin loss proposed by Yuan et al. [2020] for deep AUC maximization by setting $\ell(\cdot)$ as the square function or squared hinge function, respectively. When applied to multi-task classification (e.g., multiple classes), we can optimize the averaged AUC losses over all tasks, i.e., $AUC = \frac{1}{m}\sum_{i=1}^{m} AUC(i)$. The nested structure only comes from the term $\ell(a(\mathbf{w}) - b(\mathbf{w}))$, and we can rewrite it as the form of FCCO problem, where

$$g_i(\mathbf{w}) = \frac{1}{|\mathcal{D}_+^i|}\sum_{\mathbf{x}\in\mathcal{D}_+^i} h_{\mathbf{w}}(\mathbf{x}) - \frac{1}{|\mathcal{D}_-^i|}\sum_{\mathbf{x}\in\mathcal{D}_-^i} h_{\mathbf{w}}(\mathbf{x}), \qquad f(g_i(\mathbf{w})) = \ell(g_i(\mathbf{w})).$$

where $\mathcal{D}_{+/-}^i$ denotes the positive/negative datasets of the $i$-th task.

**Configurations.** In the experiment, we follow the setup in Zhu et al. [2022] and set the surrogate function $\ell$ as squared hinge $\ell(x) = \frac{1}{2}(\max\{c + x, 0\})^2$. We use ResNet18 as backbone network, and train

on six datasets: STL10 [Coates et al., 2011], CIFAR10 [Krizhevsky, 2009], CIFAR100 [Krizhevsky, 2009], MNIST [LeCun et al., 1998], Fashion-MNIST [Xiao et al., 2017], and SVHN [Netzer et al., 2011]. We compare our methods with previous SOTA algorithm SOX [Wang and Yang, 2022]. For our methods, parameters $\alpha$ and $\beta$ are searched from $\{0.1, 0.5, 0.9, 1.0\}$. For SOX algorithm, its parameters $\beta$ and $\gamma$ are searched from the same set. $B_1$ is set as 50 for CIFAR100 and 5 for other datasets. Inner batch size $B_2$ is chosen as 128 for all methods. We tune the learning rate from the set $\{1e - 4, 1e - 3, 2e - 3, 5e - 3, 1e - 2\}$ and pick the best one for each method. The experiments are conducted on single NVIDIA Tesla M40 GPU.

**Results.** Figure 1 shows the loss against the number of samples drawn by different methods, and all curves are averaged over 5 runs. We observe that MSVR-V1 is better than SOX on the CIFAR100 dataset, and close to it on other datasets. MSVR-v2 converges faster than SOX and MSVR-v1, and the loss of MSVR-v3 decreases most rapidly, demonstrating a low sample complexity.

# 6    Conclusion and Future Work

In this paper, we develop a novel MSVR estimator for tracking multiple functional mappings by probing only $\mathcal{O}(1)$ blocks. Equipped with this estimator, we design three algorithms for FCCO problems and obtain improved complexities across a spectrum of settings. Experimental results on multi-task deep AUC maximization also verify the effectiveness of our methods. In future work, we will investigate other applications that can be solved by using the proposed estimator.

## Acknowledgments and Disclosure of Funding

W. Jiang, Y. Wang and L. Zhang were partially supported by NSFC (62122037, 61921006). G. Li and T. Yang were partially supported by Amazon research award. The authors would like to thank the anonymous reviewers for their helpful comment.

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
