# A  Analysis

## A.1  Proof of Lemma 1

**Proof** *Denote* $\bar{\mathbf{u}}_t^i = (1 - \beta_{t+1})\mathbf{u}_t^i + \beta_{t+1}g_i(\mathbf{w}_{t+1}; \xi_{t+1}^i) + \gamma_{t+1}\left(g_i(\mathbf{w}_{t+1}; \xi_{t+1}^i) - g_i(\mathbf{w}_t; \xi_{t+1}^i)\right).$

$$\mathbb{E}\left[\left\|\mathbf{u}_{t+1}^i - g_i(\mathbf{w}_{t+1})\right\|^2\right] = \mathbb{E}\left[(1 - \frac{B_1}{m})\left\|\mathbf{u}_t^i - g_i(\mathbf{w}_{t+1})\right\|^2 + \frac{B_1}{m}\left\|\bar{\mathbf{u}}_t^i - g_i(\mathbf{w}_{t+1})\right\|^2\right] \quad (11)$$

*For the first term, we can decompose it as:*

$$
\begin{aligned}
&(1 - \frac{B_1}{m})\mathbb{E}\left[\left\|\mathbf{u}_t^i - g_i(\mathbf{w}_{t+1})\right\|^2\right] \\
=&(1 - \frac{B_1}{m})\mathbb{E}\left[\left\|\mathbf{u}_t^i - g_i(\mathbf{w}_t)\right\|^2\right] + (1 - \frac{B_1}{m})\mathbb{E}\left[\left\|g_i(\mathbf{w}_t) - g_i(\mathbf{w}_{t+1})\right\|^2\right] \\
&+ \underbrace{2(1 - \frac{B_1}{m})\mathbb{E}\left[\left(\mathbf{u}_t^i - g_i(\mathbf{w}_t)\right)\left(g_i(\mathbf{w}_t) - g_i(\mathbf{w}_{t+1})\right)\right]}_{\textcircled{1}}
\end{aligned}
\quad (12)
$$

*Also, the second term can be written as:*

$$
\begin{aligned}
&\frac{B_1}{m}\mathbb{E}\left[\left\|\bar{\mathbf{u}}_t^i - g_i(\mathbf{w}_{t+1})\right\|^2\right] \\
=&\frac{B_1}{m}\mathbb{E}\Big[\big\|(1 - \beta_{t+1})\left(\mathbf{u}_t^i - g_i(\mathbf{w}_t)\right) + (1 - \beta_{t+1})\left(g_i(\mathbf{w}_t) - g_i(\mathbf{w}_{t+1})\right) \\
&+\beta_{t+1}\left(g_i(\mathbf{w}_{t+1}; \xi_{t+1}^i) - g_i(\mathbf{w}_{t+1})\right) + \gamma_{t+1}\left(g_i(\mathbf{w}_{t+1}; \xi_{t+1}^i) - g_i(\mathbf{w}_t; \xi_{t+1}^i)\right)\big\|^2\Big] \\
=&\frac{B_1}{m}\mathbb{E}\Big[\big\|(1 - \beta_{t+1})\left(\mathbf{u}_t^i - g_i(\mathbf{w}_t)\right) + (1 - \beta_{t+1})\left(g_i(\mathbf{w}_t) - g_i(\mathbf{w}_{t+1})\right) \\
&+\gamma_{t+1}\left(g_i(\mathbf{w}_{t+1}; \xi_{t+1}^i) - g_i(\mathbf{w}_t; \xi_{t+1}^i)\right)\big\|^2\Big] + \frac{B_1\beta_{t+1}^2}{m}\mathbb{E}\left[\left\|g_i(\mathbf{w}_{t+1}; \xi_{t+1}^i) - g_i(\mathbf{w}_{t+1})\right\|^2\right] \\
&+\frac{2B_1\beta_{t+1}\gamma_{t+1}}{m}\mathbb{E}\left[\left(g_i(\mathbf{w}_{t+1}; \xi_{t+1}^i) - g_i(\mathbf{w}_{t+1})\right)\left(g_i(\mathbf{w}_{t+1}; \xi_{t+1}^i) - g_i(\mathbf{w}_t; \xi_{t+1}^i)\right)\right] \\
=&\frac{B_1}{m}\mathbb{E}\left[(1 - \beta_{t+1})^2\left\|\left(\mathbf{u}_t^i - g_i(\mathbf{w}_t)\right)\right\|^2\right] \\
&+\frac{B_1}{m}\mathbb{E}\left[\left\|(1 - \beta_{t+1})\left(g_i(\mathbf{w}_t) - g_i(\mathbf{w}_{t+1})\right) + \gamma_{t+1}\left(g_i(\mathbf{w}_{t+1}; \xi_{t+1}^i) - g_i(\mathbf{w}_t; \xi_{t+1}^i)\right)\right\|^2\right] \\
&+\underbrace{\frac{2B_1}{m}(1 - \beta_{t+1})(1 - \beta_{t+1} - \gamma_{t+1})\mathbb{E}\left[\left(\mathbf{u}_t^i - g_i(\mathbf{w}_t)\right)\left(g_i(\mathbf{w}_t) - g_i(\mathbf{w}_{t+1})\right)\right]}_{\textcircled{2}} \\
&+\frac{B_1\beta_{t+1}^2}{m}\mathbb{E}\left[\left\|\left(g_i(\mathbf{w}_{t+1}; \xi_{t+1}^i) - g_i(\mathbf{w}_{t+1})\right)\right\|^2\right] \\
&+\frac{2B_1\beta_{t+1}\gamma_{t+1}}{m}\mathbb{E}\left[\left(g_i(\mathbf{w}_{t+1}; \xi_{t+1}^i) - g_i(\mathbf{w}_{t+1})\right)\left(g_i(\mathbf{w}_{t+1}; \xi_{t+1}^i) - g_i(\mathbf{w}_t; \xi_{t+1}^i)\right)\right]
\end{aligned}
$$
$$(13)$$

*The second equation is because of* $\mathbb{E}\left[g_i(\mathbf{w}_{t+1}; \xi_{t+1}^i) - g_i(\mathbf{w}_{t+1})\right] = 0$. *The last equation is due to* $\mathbb{E}\left[g_i(\mathbf{w}_{t+1}; \xi_{t+1}^i) - g_i(\mathbf{w}_t; \xi_{t+1}^i)\right] = g_i(\mathbf{w}_{t+1}) - g_i(\mathbf{w}_t)$ *and* $\mathbf{u}_t^i$ *is independent of* $\xi_{t+1}^i$. *We want to ensure* $\textcircled{1} + \textcircled{2} = 0$, *which requires that* $2(1 - \frac{B_1}{m}) + \frac{2B_1}{m}(1 - \beta_{t+1})(1 - \beta_{t+1} - \gamma_{t+1}) = 0$. *Solve* $\gamma_{t+1}$ *and we have* $\gamma_{t+1} = \frac{m - B_1}{B_1(1 - \beta_{t+1})} + (1 - \beta_{t+1})$. *According to equation (12) and equation (13),*

*the equation (11) can now be written as:*

$$\mathbb{E}\left[\left\|\mathbf{u}_{t+1}^i - g_i(\mathbf{w}_{t+1})\right\|^2\right]$$

$$=\mathbb{E}\bigg[\left(1 - \frac{B_1}{m} + \frac{(1-\beta_{t+1})^2 B_1}{m}\right)\left\|\mathbf{u}_t^i - g_i(\mathbf{w}_t)\right\|^2 + (1 - \frac{B_1}{m})\left\|g_i(\mathbf{w}_t) - g_i(\mathbf{w}_{t+1})\right\|^2$$

$$+ \frac{B_1\beta_{t+1}^2}{m}\left\|\left(g_i(\mathbf{w}_{t+1};\xi_{t+1}^i) - g_i(\mathbf{w}_{t+1})\right)\right\|^2$$

$$+ \frac{B_1}{m}\left\|(1-\beta_{t+1})\left(g_i(\mathbf{w}_t) - g_i(\mathbf{w}_{t+1})\right) + \gamma_{t+1}\left(g_i(\mathbf{w}_{t+1};\xi_{t+1}^i) - g_i(\mathbf{w}_t;\xi_{t+1}^i)\right)\right\|^2$$

$$+ \frac{2B_1\beta_{t+1}\gamma_{t+1}}{m}\left(g_i(\mathbf{w}_{t+1};\xi_{t+1}^i) - g_i(\mathbf{w}_{t+1})\right)\left(g_i(\mathbf{w}_{t+1};\xi_{t+1}^i) - g_i(\mathbf{w}_t;\xi_{t+1}^i)\right)\bigg]$$

$$=\mathbb{E}\bigg[\left(1 - \frac{B_1}{m} + \frac{(1-\beta_{t+1})^2 B_1}{m}\right)\left\|\mathbf{u}_t^i - g_i(\mathbf{w}_t)\right\|^2 + (1 - \frac{B_1}{m})\left\|g_i(\mathbf{w}_t) - g_i(\mathbf{w}_{t+1})\right\|^2$$

$$+ \frac{B_1\beta_{t+1}^2}{m}\left\|\left(g_i(\mathbf{w}_{t+1};\xi_{t+1}^i) - g_i(\mathbf{w}_{t+1})\right)\right\|^2 + \frac{B_1(1-\beta_{t+1})^2}{m}\left\|\left(g_i(\mathbf{w}_t) - g_i(\mathbf{w}_{t+1})\right)\right\|^2$$

$$- \frac{2B_1(1-\beta_{t+1})}{m}\gamma_{t+1}\left\|\left(g_i(\mathbf{w}_t) - g_i(\mathbf{w}_{t+1})\right)\right\|^2 \tag{14}$$

$$+ \frac{B_1\gamma_{t+1}^2}{m}\left\|\left(g_i(\mathbf{w}_{t+1};\xi_{t+1}^i) - g_i(\mathbf{w}_t;\xi_{t+1}^i)\right)\right\|^2$$

$$+ \frac{2B_1\beta_{t+1}\gamma_{t+1}}{m}\left(g_i(\mathbf{w}_{t+1};\xi_{t+1}^i) - g_i(\mathbf{w}_{t+1})\right)\left(g_i(\mathbf{w}_{t+1};\xi_{t+1}^i) - g_i(\mathbf{w}_t;\xi_{t+1}^i)\right)\bigg]$$

$$\leq \left(1 - \frac{\beta_{t+1}B_1}{m}\right)\mathbb{E}\left[\left\|\mathbf{u}_t^i - g_i(\mathbf{w}_t)\right\|^2\right] + \frac{B_1\beta_{t+1}^2}{m}\mathbb{E}\left[\left\|\left(g_i(\mathbf{w}_{t+1};\xi_{t+1}^i) - g_i(\mathbf{w}_{t+1})\right)\right\|^2\right]$$

$$+ \frac{4mC_g^2}{B_1}\mathbb{E}\left[\left\|\mathbf{w}_{t+1} - \mathbf{w}_t\right\|^2\right]$$

$$+ \frac{2B_1\beta_{t+1}\gamma_{t+1}}{m}\mathbb{E}\left[\left(g_i(\mathbf{w}_{t+1};\xi_{t+1}^i) - g_i(\mathbf{w}_{t+1})\right)\left(g_i(\mathbf{w}_{t+1};\xi_{t+1}^i) - g_i(\mathbf{w}_t;\xi_{t+1}^i)\right)\right]$$

$$\leq \left(1 - \frac{\beta_{t+1}B_1}{m}\right)\mathbb{E}\left[\left\|\mathbf{u}_t^i - g_i(\mathbf{w}_t)\right\|^2\right] + \frac{2B_1\beta_{t+1}^2}{m}\mathbb{E}\left[\left\|\left(g_i(\mathbf{w}_{t+1};\xi_{t+1}^i) - g_i(\mathbf{w}_{t+1})\right)\right\|^2\right]$$

$$+ \frac{8mC_g^2}{B_1}\mathbb{E}\left[\left\|\mathbf{w}_{t+1} - \mathbf{w}_t\right\|^2\right]$$

*The second equation is due to*

$$\mathbb{E}\left[\left(g_i(\mathbf{w}_{t+1};\xi_{t+1}^i) - g_i(\mathbf{w}_t;\xi_{t+1}^i)\right)\left(g_i(\mathbf{w}_{t+1}) - g_i(\mathbf{w}_t)\right)\right] = \left\|g_i(\mathbf{w}_{t+1}) - g_i(\mathbf{w}_t)\right\|^2.$$

*The first inequality is because of $\gamma_{t+1} \leq \frac{2m}{B_1}$ (since $\beta_{t+1} \leq \frac{1}{2}$) and $1 - \frac{B_1}{m} + \frac{B_1}{m}(1-\beta_{t+1})^2 \leq 2\frac{B_1}{m}(1-\beta_{t+1})\gamma_{t+1}$. The last inequality is due to the fact that*

$$\frac{2B_1\beta_{t+1}\gamma_{t+1}}{m}\mathbb{E}\left[\left(g_i(\mathbf{w}_{t+1};\xi_{t+1}^i) - g_i(\mathbf{w}_{t+1})\right)\left(g_i(\mathbf{w}_{t+1};\xi_{t+1}^i) - g_i(\mathbf{w}_t;\xi_{t+1}^i)\right)\right]$$

$$\leq \frac{2B_1\beta_{t+1}\gamma_{t+1}}{m}\mathbb{E}\left[\left\|g_i(\mathbf{w}_{t+1};\xi_{t+1}^i) - g_i(\mathbf{w}_{t+1})\right\|\left\|g_i(\mathbf{w}_{t+1};\xi_{t+1}^i) - g_i(\mathbf{w}_t;\xi_{t+1}^i)\right\|\right]$$

$$\leq \frac{B_1\beta_{t+1}^2}{m}\mathbb{E}\left[\left\|g_i(\mathbf{w}_{t+1};\xi_{t+1}^i) - g_i(\mathbf{w}_{t+1})\right\|^2\right] + \frac{B_1\gamma_{t+1}^2}{m}\mathbb{E}\left[\left\|g_i(\mathbf{w}_{t+1};\xi_{t+1}^i) - g_i(\mathbf{w}_t;\xi_{t+1}^i)\right\|^2\right]$$

$$\leq \frac{B_1\beta_{t+1}^2}{m}\mathbb{E}\left[\left\|g_i(\mathbf{w}_{t+1};\xi_{t+1}^i) - g_i(\mathbf{w}_{t+1})\right\|^2\right] + \frac{4mC_g^2}{B_1}\mathbb{E}\left[\left\|\mathbf{w}_{t+1} - \mathbf{w}_t\right\|^2\right]$$

*Finally, we have:*

$$\mathbb{E}\left[\left\|\mathbf{u}_{t+1} - g(\mathbf{w}_{t+1})\right\|^2\right] = \sum_{i=1}^m \mathbb{E}\left[\left\|\mathbf{u}_{t+1}^i - g_i(\mathbf{w}_{t+1})\right\|^2\right]$$

$$\leq \left(1 - \frac{\beta_{t+1}B_1}{m}\right)\mathbb{E}\left[\left\|\mathbf{u}_t - g(\mathbf{w}_t)\right\|^2\right] + \frac{8m^2C_g^2}{B_1}\mathbb{E}\left[\left\|\mathbf{w}_{t+1} - \mathbf{w}_t\right\|^2\right] + \frac{2B_1\sigma^2\beta_{t+1}^2}{B_2}$$

## A.2 Proof of Lemma 2

**Proof** *We will start with single block and them sum over multiple blocks. To start, we have*

$$\left\| \mathbf{u}_{t+1}^i - g_i(\mathbf{w}_{t+1}) \right\|^2$$

$$= \left( 1 - \frac{B_1}{m} \right) \left\| \mathbf{u}_t^i - g_i(\mathbf{w}_{t+1}) \right\|^2$$

$$+ \frac{B_1}{m} \left\| (1 - \beta_{t+1})\mathbf{u}_t^i + \beta_{t+1} g_i(\mathbf{w}_{t+1}; \xi_{t+1}^i) + \gamma_{t+1} \nabla g_i(\mathbf{w}_{t+1}; \xi_{t+1}^i)^\top (\mathbf{w}_{t+1} - \mathbf{w}_t) - g_i(\mathbf{w}_{t+1}) \right\|^2$$

$$= \left( 1 - \frac{B_1}{m} \right) \left( \left\| \mathbf{u}_t^i - g_i(\mathbf{w}_t) \right\|^2 + \underbrace{2(\mathbf{u}_t^i - g_i(\mathbf{w}_t))^\top (g_i(\mathbf{w}_t) - g_i(\mathbf{w}_{t+1}))}_{A_0} + \left\| g_i(\mathbf{w}_t) - g_i(\mathbf{w}_{t+1}) \right\|^2 \right)$$

$$+ \frac{B_1}{m} \underbrace{\left\| (1 - \beta_{t+1})\mathbf{u}_t^i + \beta_{t+1} g_i(\mathbf{w}_{t+1}; \xi_{t+1}^i) + \gamma_{t+1} \nabla g_i(\mathbf{w}_{t+1}; \xi_{t+1}^i)^\top (\mathbf{w}_{t+1} - \mathbf{w}_t) - g_i(\mathbf{w}_{t+1}) \right\|^2}_{A}$$

*Next, we will proceed to decompose A.*

$$A = \left\| (1 - \beta_{t+1})(\mathbf{u}_t^i - g_i(\mathbf{w}_t)) + (1 - \beta_{t+1})(g_i(\mathbf{w}_t) - g_i(\mathbf{w}_{t+1})) \right.$$

$$+ \beta_{t+1}(g_i(\mathbf{w}_{t+1}; \xi_{t+1}^i) - g_i(\mathbf{w}_{t+1})) + \gamma_{t+1}(g_i(\mathbf{w}_{t+1}) - g_i(\mathbf{w}_t))$$

$$\left. + \gamma_{t+1}(g_i(\mathbf{w}_t) - g_i(\mathbf{w}_{t+1})) + \nabla g_i(\mathbf{w}_{t+1}; \xi_{t+1}^i)^\top (\mathbf{w}_{t+1} - \mathbf{w}_t) \right\|^2$$

$$= \left\| (1 - \beta_{t+1})(\mathbf{u}_t^i - g_i(\mathbf{w}_t)) + (\gamma_{t+1} + \beta_{t+1} - 1)(g_i(\mathbf{w}_{t+1}) - g_i(\mathbf{w}_t)) \right.$$

$$+ \gamma_{t+1} \underbrace{(g_i(\mathbf{w}_t) - g_i(\mathbf{w}_{t+1}) + \nabla g_i(\mathbf{w}_{t+1})^\top (\mathbf{w}_{t+1} - \mathbf{w}_t))}_{\Delta_t}$$

$$+ \beta_{t+1}(g_i(\mathbf{w}_{t+1}; \xi_{t+1}^i) - g_i(\mathbf{w}_{t+1}))$$

$$\left. + \gamma_{t+1}(\nabla g_i(\mathbf{w}_{t+1}; \xi_{t+1}^i) - \nabla g_i(\mathbf{w}_{t+1}))^\top (\mathbf{w}_{t+1} - \mathbf{w}_t) \right\|^2$$

*By taking expectation over A, we have*

$$\mathbb{E}[A] = \mathbb{E}\left[ (1 - \beta_{t+1})^2 \|\mathbf{u}_t^i - g_i(\mathbf{w}_t)\|^2 + \gamma_{t+1}^2 \|\Delta_t\|^2 + (\gamma_{t+1} + \beta_{t+1} - 1)^2 \|g_i(\mathbf{w}_{t+1}) - g_i(\mathbf{w}_t)\|^2 \right.$$

$$+ \underbrace{2(1 - \beta_{t+1})(\gamma_{t+1} + \beta_{t+1} - 1)(\mathbf{u}_t^i - g_i(\mathbf{w}_t)^\top (g_i(\mathbf{w}_{t+1}) - g_i(\mathbf{w}_t))}_{A_1}$$

$$+ 2\gamma_{t+1}(\gamma_{t+1} + \beta_{t+1} - 1)\Delta_t^\top (g_i(\mathbf{w}_{t+1}) - g_i(\mathbf{w}_t))$$

$$\left. + 2(1 - \beta_{t+1})\gamma_{t+1}(\mathbf{u}_t^i - g_i(\mathbf{w}_t))^\top \Delta_t + \frac{2\beta_{t+1}^2 \sigma^2}{B_2} + \frac{2\gamma_{t+1}^2 \sigma^2}{B_2} \|\mathbf{w}_{t+1} - \mathbf{w}_t\|^2 \right]$$

*Since $\gamma_{t+1} + \beta_{t+1} - 1 = \frac{m - B_1}{B_1(1 - \beta_{t+1})}$, the terms involving $A_0, A_1$ will cancel. As a result, we have*

$$\mathbb{E}\left[ \left\| \mathbf{u}_{t+1}^i - g_i(\mathbf{w}_{t+1}) \right\|^2 \right]$$

$$= \mathbb{E}\left[ \left( 1 - \frac{B_1}{m} \right) \left\| \mathbf{u}_t^i - g_i(\mathbf{w}_t) \right\|^2 + \left\| g_i(\mathbf{w}_t) - g_i(\mathbf{w}_{t+1}) \right\|^2 \right.$$

$$+ \frac{B_1}{m}(1 - \beta_{t+1})^2(1 + \beta_{t+1})\|\mathbf{u}_t^i - g_i(\mathbf{w}_t)\|^2 + (2 + \frac{1}{\beta_{t+1}})\gamma_{t+1}^2 \|\Delta_t\|^2$$

$$\left. + 2(\gamma_{t+1} + \beta_{t+1} - 1)^2 \|g_i(\mathbf{w}_{t+1}) - g_i(\mathbf{w}_t)\|^2 + \frac{2\beta_{t+1}^2 \sigma^2}{B_2} + \frac{2\gamma_{t+1}^2 \sigma^2}{B_2} \|\mathbf{w}_{t+1} - \mathbf{w}_t\|^2 \right]$$

*Since $\|\Delta_t\| \leq \min(\frac{L_g}{2}\|\mathbf{w}_{t+1} - \mathbf{w}_t\|^2, 2C_g\|\mathbf{w}_{t+1} - \mathbf{w}_t\|)$ and $\|g_i(\mathbf{w}_{t+1}) - g_i(\mathbf{w}_t)\| \leq C_g\|\mathbf{w}_{t+1} - \mathbf{w}_t\|^2$, we have*

$$\mathbb{E}\left[\left\|\mathbf{u}_{t+1}^i - g_i(\mathbf{w}_{t+1})\right\|^2\right]$$

$$=\mathbb{E}\Bigg[\left(1 - \frac{B_1}{m}\right)(\mathbf{u}_t^i - g_i(\mathbf{w}_t))^2 + C_g^2\|\mathbf{w}_{t+1} - \mathbf{w}_t\|^2$$

$$+ \frac{B_1}{m}(1 - \beta_{t+1})\|\mathbf{u}_t^i - g_i(\mathbf{w}_t)\|^2 + (2 + \frac{1}{\beta_{t+1}})\gamma_{t+1}^2 \frac{L_g^2\|\mathbf{w}_{t+1} - \mathbf{w}_t\|^2}{4}\|\mathbf{w}_{t+1} - \mathbf{w}_t\|^2$$

$$+ 2(\gamma_{t+1} + \beta_{t+1} - 1)^2 C_g^2\|\mathbf{w}_{t+1} - \mathbf{w}_t\|^2 + \frac{2\beta_{t+1}^2\sigma^2}{B_2} + \frac{2\gamma_{t+1}^2\sigma^2}{B_2}\|\mathbf{w}_{t+1} - \mathbf{w}_t\|^2\Bigg]$$

*Note that we have $\|\mathbf{w}_{t+1} - \mathbf{w}_t\|^2 \leq \eta_{t+1}^2 C_F^2$, $\gamma_{t+1} \leq \frac{2m}{B_1}$ and $\eta_{t+1} \leq \sqrt{\beta_{t+1}}$. Therefore*

$$\mathbb{E}\left[\left\|\mathbf{u}_{t+1}^i - g_i(\mathbf{w}_{t+1})\right\|^2\right]$$

$$\leq \left(1 - \frac{B_1\beta_{t+1}}{m}\right)\mathbb{E}\left[\left\|\mathbf{u}_t^i - g_i(\mathbf{w}_t)\right\|^2\right] + \frac{2B_1\beta_{t+1}^2\sigma^2}{mB_2}$$

$$+ \mathbb{E}\left[\frac{4mL_g^2}{\beta_{t+1}B_1}\|\mathbf{w}_{t+1} - \mathbf{w}_t\|^4 + \frac{9mC_g^2}{B_1}\|\mathbf{w}_{t+1} - \mathbf{w}_t\|^2 + \frac{8m\sigma^2}{B_1B_2}\|\mathbf{w}_{t+1} - \mathbf{w}_t\|^2\right]$$

$$\leq \left(1 - \frac{B_1\beta_{t+1}}{m}\right)\mathbb{E}\left[\left\|\mathbf{u}_t^i - g_i(\mathbf{w}_t)\right\|^2\right] + \frac{2B_1\beta_{t+1}^2\sigma^2}{mB_2}$$

$$+ \left(4L_g^2 C_F^2 + 9C_g^2 + \frac{8\sigma^2}{B_2}\right)\frac{m}{B_1}\mathbb{E}\left[\|\mathbf{w}_{t+1} - \mathbf{w}_t\|^2\right]$$

*Finally, we have:*

$$\mathbb{E}[\|\mathbf{u}_{t+1} - g(\mathbf{w}_{t+1})\|^2] \leq \left(1 - \frac{B_1\beta_{t+1}}{m}\right)\mathbb{E}\left[\|\mathbf{u}_t - g(\mathbf{w}_t)\|^2\right] + \frac{2B_1\beta_{t+1}^2\sigma^2}{B_2}$$

$$+ \left(4L_g^2 C_F^2 + 9C_g^2 + \frac{8\sigma^2}{B_2}\right)\frac{m^2}{B_1}\mathbb{E}\left[\|\mathbf{w}_{t+1} - \mathbf{w}_t\|^2\right]$$

### A.3   Proof of Lemma 3

**Proof** *Note that we have:*

$$\mathbb{E}\left[\left\|\widehat{g}_i(\mathbf{w}_{t+1};\xi_{t+1}^i) - g_i(\mathbf{w}_{t+1})\right\|^2\right]$$

$$=\mathbb{E}\left[\left\|g_i(\mathbf{w}_{t+1};\xi_{t+1}^i) - g_i(\mathbf{w}_\tau;\xi_{t+1}^i) + g_i(\mathbf{w}_\tau) - g_i(\mathbf{w}_{t+1})\right\|^2\right]$$

$$=\mathbb{E}\left[\left\|g_i(\mathbf{w}_{t+1};\xi_{t+1}^i) - g_i(\mathbf{w}_\tau;\xi_{t+1}^i)\right\|^2\right] + \|g_i(\mathbf{w}_\tau) - g_i(\mathbf{w}_{t+1})\|^2$$

$$+ 2\mathbb{E}\left[g_i(\mathbf{w}_{t+1};\xi_{t+1}^i) - g_i(\mathbf{w}_\tau;\xi_{t+1}^i)\right]\left[g_i(\mathbf{w}_\tau) - g_i(\mathbf{w}_{t+1})\right] \tag{15}$$

$$=\mathbb{E}\left[\left\|g_i(\mathbf{w}_{t+1};\xi_{t+1}^i) - g_i(\mathbf{w}_\tau;\xi_{t+1}^i)\right\|^2\right] - \|g_i(\mathbf{w}_\tau) - g_i(\mathbf{w}_{t+1})\|^2$$

$$\leq C_g^2\|\mathbf{w}_{t+1} - \mathbf{w}_\tau\|^2$$

*Since $\tau$ is the closest small index such that $\tau \mod I = 0$, we have:*

$$\sum_{t=1}^T \|\mathbf{w}_{t+1} - \mathbf{w}_\tau\|^2 \leq \sum_{t=1}^T \left\|\sum_{k=\tau}^t (\mathbf{w}_{k+1} - \mathbf{w}_k)\right\|^2$$

$$\leq \sum_{t=1}^T \sum_{k=\tau}^t I\|\mathbf{w}_{k+1} - \mathbf{w}_k\|^2 \leq I^2\sum_{t=1}^T \|\mathbf{w}_{t+1} - \mathbf{w}_t\|^2 \tag{16}$$

*We can then apply the same analysis as in Section A.1, until equation (14):*

$$
\mathbb{E}\left[\left\|\mathbf{u}_{t+1}^i - g_i(\mathbf{w}_{t+1})\right\|^2\right]
$$

$$
\leq \left(1 - \frac{\beta B_1}{m}\right)\mathbb{E}\left[\left\|\mathbf{u}_t^i - g_i(\mathbf{w}_t)\right\|^2\right] + \frac{2B_1\beta^2}{m}\mathbb{E}\left[\left\|\left(\widehat{g}_i(\mathbf{w}_{t+1}; \xi_{t+1}^i) - g_i(\mathbf{w}_{t+1})\right)\right\|^2\right]
$$

$$
+ \frac{8mC_g^2}{B_1}\mathbb{E}\left[\left\|\mathbf{w}_{t+1} - \mathbf{w}_t\right\|^2\right]
$$

$$
\leq \left(1 - \frac{\beta B_1}{m}\right)\mathbb{E}\left[\left\|\mathbf{u}_t^i - g_i(\mathbf{w}_t)\right\|^2\right] + \left(\frac{2B_1C_g^2\beta^2 I^2}{m} + \frac{8mC_g^2}{B_1}\right)\mathbb{E}\left[\left\|\mathbf{w}_{t+1} - \mathbf{w}_t\right\|^2\right]
$$

$$
\leq \left(1 - \frac{\beta B_1}{m}\right)\mathbb{E}\left[\left\|\mathbf{u}_t^i - g_i(\mathbf{w}_t)\right\|^2\right] + \frac{10mC_g^2}{B_1}\mathbb{E}\left[\left\|\mathbf{w}_{t+1} - \mathbf{w}_t\right\|^2\right]
$$

*The last inequality is due to $\beta I \leq \frac{m}{B_1}$. Finally, we have:*

$$
\mathbb{E}\left[\left\|\mathbf{u}_{t+1} - g(\mathbf{w}_{t+1})\right\|^2\right] = \sum_{i=1}^m \mathbb{E}\left[\left\|\mathbf{u}_{t+1}^i - g_i(\mathbf{w}_{t+1})\right\|^2\right]
$$

$$
\leq (1 - \frac{B_1\beta}{m})\mathbb{E}\left[\left\|\mathbf{u}_t - g(\mathbf{w}_t)\right\|^2\right] + \frac{10m^2C_g^2}{B_1}\left\|\mathbf{w}_{t+1} - \mathbf{w}_t\right\|^2
$$

## A.4   Proof of Theorem 1

We denote constant $C = \max\left\{1, C_g^2, L_F^2, C_F^2, \sigma^2, L_f^2C_g^2, L_g^2C_f^2, L_f^2C_g^4, L_f^2C_g^2\sigma^2, C_f^2(\sigma^2 + C_g^2)\right\}$.

**Lemma 4** *(Lemma 2 in Li et al. [2021]) Suppose function $F$ is $L_F$-smooth and consider the update $\mathbf{w}_{t+1} := \mathbf{w}_t - \eta_t\mathbf{z}_t$. With $\eta_t L \leq \frac{1}{2}$, we have:*

$$
F(\mathbf{w}_{t+1}) \leq F(\mathbf{w}_t) - \frac{\eta_t}{2}\|\nabla F(\mathbf{w}_t)\|^2 + \frac{\eta_t}{2}\|\mathbf{z}_t - \nabla F(\mathbf{w}_t)\|^2 - \frac{\eta_t}{4}\|\mathbf{z}_t\|^2
$$

**Lemma 5** *Denote $\|\mathbf{u}_t - g(\mathbf{w}_t)\|^2 = \sum_{i=1}^m \|\mathbf{u}_t^i - g_i(\mathbf{w}_t)\|^2$ and $\|\mathbf{u}_t - \mathbf{u}_{t-1}\|^2 = \sum_{i=1}^m \|\mathbf{u}_t^i - \mathbf{u}_{t-1}^i\|^2$.*

$$
\mathbb{E}\left[\|\mathbf{z}_{t+1} - \nabla F(\mathbf{w}_{t+1})\|^2\right] \leq (1 - \alpha_{t+1})\mathbb{E}\left[\|\mathbf{z}_t - \nabla F(\mathbf{w}_t)\|^2\right] + \frac{3C\eta_t^2\mathbb{E}\left[\|\mathbf{z}_t\|^2\right]}{\alpha_{t+1}}
$$

$$
+ \frac{4L_f^2C_g^2}{m}\mathbb{E}\left[\|\mathbf{u}_{t+1} - \mathbf{u}_t\|^2\right] + \frac{2\alpha_{t+1}^2C_f^2(\sigma^2 + C_g^2)}{\min\{B_1, B_2\}} + \frac{5\alpha_{t+1}L_f^2C_g^2}{m}\mathbb{E}\left[\|\mathbf{u}_t - g(\mathbf{w}_t)\|^2\right]
$$

**Proof** *According to Lemma 1 in Wang and Yang [2022], if $\alpha \leq \frac{2}{7}$, we have:*

$$
\mathbb{E}\left[\|\mathbf{z}_{t+1} - \nabla F(\mathbf{w}_{t+1})\|^2\right] \leq (1 - \alpha_{t+1})\mathbb{E}\left[\|\mathbf{z}_t - \nabla F(\mathbf{w}_t)\|^2\right] + \frac{2L_F^2\eta_t^2\mathbb{E}\left[\|\mathbf{z}_t\|^2\right]}{\alpha_{t+1}}
$$

$$
+ \frac{3L_f^2C_g^2}{m}\mathbb{E}\left[\|\mathbf{u}_{t+1} - \mathbf{u}_t\|^2\right] + \frac{2\alpha_{t+1}^2C_f^2(\sigma^2 + C_g^2)}{\min\{B_1, B_2\}} + \frac{5\alpha_{t+1}L_f^2C_g^2}{m}\mathbb{E}\left[\|\mathbf{u}_{t+1} - g(\mathbf{w}_{t+1})\|^2\right].
$$

*By setting $\alpha \leq \frac{1}{15}$, we have the above lemma.*

**Lemma 6** *If $\beta_{t+1} \leq \frac{1}{2}$, we have:*

$$
\mathbb{E}\left[\|\mathbf{u}_{t+1} - \mathbf{u}_t\|^2\right] \leq \frac{2B_1\beta_{t+1}^2\sigma^2}{B_2} + \frac{4B_1\beta_{t+1}^2}{m}\mathbb{E}\left[\|\mathbf{u}_t - g(\mathbf{w}_t)\|^2\right] + \frac{9m^2C_g^2}{B_1}\mathbb{E}\left[\|\mathbf{w}_{t+1} - \mathbf{w}_t\|^2\right]
$$

**Proof** *Note that with $\beta_{t+1} \leq \frac{1}{2}$, we have $\gamma_{t+1} \leq \frac{2m}{B_1}$*

$$\mathbb{E}\left[\|\mathbf{u}_{t+1} - \mathbf{u}_t\|^2\right]$$

$$=\frac{B_1}{m}\sum_{i=1}^{m}\mathbb{E}\left[\left\|\beta_{t+1}\left(g_i(\mathbf{w}_{t+1};\xi_{t+1}^i) - \mathbf{u}_t^i\right) + \gamma_{t+1}\left(g_i(\mathbf{w}_{t+1};\xi_{t+1}^i) - g_i(\mathbf{w}_t;\xi_{t+1}^i)\right)\right\|^2\right]$$

$$\leq\frac{B_1}{m}\sum_{i=1}^{m}\mathbb{E}\left[2\beta_{t+1}^2\left\|g_i(\mathbf{w}_{t+1};\xi_{t+1}^i) - \mathbf{u}_t^i\right\|^2 + 2\gamma_{t+1}^2\|g_i(\mathbf{w}_{t+1};\xi_{t+1}^i) - g_i(\mathbf{w}_t;\xi_{t+1}^i)\|^2\right]$$

$$\leq\mathbb{E}\left[\frac{2B_1\beta_{t+1}^2}{m}\sum_{i=1}^{m}\left\|g_i(\mathbf{w}_{t+1};\xi_{t+1}^i) - \mathbf{u}_t^i\right\|^2 + 2B_1\gamma_{t+1}^2 C_g^2\|\mathbf{w}_{t+1} - \mathbf{w}_t\|^2\right]$$

$$\leq\frac{2B_1\beta_{t+1}^2}{m}\sum_{i=1}^{m}\left(\mathbb{E}\left[\left\|g_i(\mathbf{w}_{t+1};\xi_{t+1}^i) - g_i(\mathbf{w}_{t+1})\right\|^2\right] + \mathbb{E}\left[\left\|g_i(\mathbf{w}_{t+1}) - \mathbf{u}_t^i\right\|^2\right]\right)$$
$$+ 2B_1\gamma_{t+1}^2 C_g^2\mathbb{E}\left[\|\mathbf{w}_{t+1} - \mathbf{w}_t\|^2\right]$$

$$\leq\frac{2B_1\beta_{t+1}^2\sigma^2}{B_2} + \frac{2B_1\beta_{t+1}^2}{m}\sum_{i=1}^{m}\mathbb{E}\left[\left\|g_i(\mathbf{w}_{t+1}) - \mathbf{u}_t^i\right\|^2\right] + \frac{8m^2C_g^2}{B_1}\mathbb{E}\left[\|\mathbf{w}_{t+1} - \mathbf{w}_t\|^2\right]$$

$$\leq\frac{2B_1\beta_{t+1}^2\sigma^2}{B_2} + \frac{4B_1\beta_{t+1}^2}{m}\sum_{i=1}^{m}\|g_i(\mathbf{w}_{t+1}) - g_i(\mathbf{w}_t)\|^2 + \frac{4B_1\beta_{t+1}^2}{m}\sum_{i=1}^{m}\mathbb{E}\left[\left\|g_i(\mathbf{w}_t) - \mathbf{u}_t^i\right\|^2\right]$$
$$+ \frac{8m^2C_g^2}{B_1}\mathbb{E}\left[\|\mathbf{w}_{t+1} - \mathbf{w}_t\|^2\right]$$

$$\leq\frac{2B_1\beta_{t+1}^2\sigma^2}{B_2} + \frac{4B_1\beta_{t+1}^2}{m}\sum_{i=1}^{m}\mathbb{E}\left[\left\|g_i(\mathbf{w}_t) - \mathbf{u}_t^i\right\|^2\right] + \frac{9m^2C_g^2}{B_1}\mathbb{E}\left[\|\mathbf{w}_{t+1} - \mathbf{w}_t\|^2\right].$$

*The third inequality is due to $\mathbb{E}\left[g_i(\mathbf{w}_{t+1};\xi_{t+1}^i) - g_i(\mathbf{w}_{t+1})\right] = 0$.*

**The rest proof of Theorem 1** Let $\Gamma_t = F(\mathbf{w}_t) + \frac{B_1}{c_0\eta_{t-1}m^2}\|\mathbf{u}_t - g(\mathbf{w}_t)\|^2 + \frac{1}{c_0}\|\mathbf{z}_t - \nabla F(\mathbf{w}_t)\|^2$. By setting $\eta_t = \frac{2\alpha_{t+1}}{c_0}$, $C_0 = 72C$, $\eta_t \leq \frac{B_1}{4m}$ we have:

$$\mathbb{E}\left[\Gamma_{t+1} - \Gamma_t\right]$$

$$=\mathbb{E}\left[F(\mathbf{w}_{t+1}) - F(\mathbf{w}_t) + \frac{B_1}{c_0\eta_t m^2}\|\mathbf{u}_{t+1} - g(\mathbf{w}_{t+1})\|^2 + \frac{1}{c_0}\|\mathbf{z}_{t+1} - \nabla F(\mathbf{w}_{t+1})\|^2\right.$$
$$\left. - \frac{B_1}{c_0\eta_{t-1}m^2}\|\mathbf{u}_t - g(\mathbf{w}_t)\|^2 - \frac{1}{c_0}\|\mathbf{z}_t - \nabla F(\mathbf{w}_t)\|^2\right]$$

$$\leq\mathbb{E}\left[-\frac{\eta_t}{2}\|\nabla F(\mathbf{w}_t)\|^2 + \frac{\eta_t}{2}\|\mathbf{z}_t - \nabla F(\mathbf{w}_t)\|^2 - \frac{\eta_t}{4}\|\mathbf{z}_t\|^2 - \frac{\alpha_{t+1}}{c_0}\|\mathbf{z}_t - \nabla F(\mathbf{w}_t)\|^2\right.$$
$$+ \frac{3C\eta_t^2}{\alpha_{t+1}c_0}\|\mathbf{z}_t\|^2 + \frac{4L_f^2C_g^2}{mc_0}\|\mathbf{u}_{t+1} - \mathbf{u}_t\|^2 + \frac{2\alpha_{t+1}^2 C_f^2\left(\sigma^2 + C_g^2\right)}{\min\{B_1, B_2\}c_0} + \frac{8C\eta_t}{c_0}\|\mathbf{z}_t\|^2$$
$$\left. + \left(\frac{5\alpha_{t+1}L_f^2C_g^2}{mc_0} + \frac{B_1}{c_0\eta_t m^2} - \frac{B_1^2\beta_{t+1}}{m^3c_0\eta_t} - \frac{B_1}{c_0\eta_{t-1}m^2}\right)\|\mathbf{u}_t - g(\mathbf{w}_t)\|^2 + \frac{2B_1^2\beta_{t+1}^2\sigma^2}{B_2m^2c_0\eta_t}\right]$$

$$\leq\mathbb{E}\left[-\frac{\eta_t}{2}\|\nabla F(\mathbf{w}_t)\|^2 + \frac{4L_f^2C_g^2}{mc_0}\|\mathbf{u}_{t+1} - \mathbf{u}_t\|^2 + \frac{2\alpha_{t+1}^2C_f^2\left(\sigma^2 + C_g^2\right)}{\min\{B_1, B_2\}c_0} - \frac{\eta_t}{8}\|\mathbf{z}_t\|^2\right.$$
$$\left. + \left(\frac{5\alpha_{t+1}C}{mc_0} + \frac{B_1}{c_0\eta_t m^2} - \frac{B_1^2\beta_{t+1}}{m^3c_0\eta_t} - \frac{B_1}{c_0\eta_{t-1}m^2}\right)\|\mathbf{u}_t - g(\mathbf{w}_t)\|^2 + \frac{2B_1^2\beta_{t+1}^2\sigma^2}{B_2m^2c_0\eta_t}\right]$$

$$\leq\mathbb{E}\left[-\frac{\eta_t}{2}\|\nabla F(\mathbf{w}_t)\|^2 + \frac{2\alpha_{t+1}^2C}{\min\{B_1, B_2\}c_0} + \frac{4B_1^2\beta_{t+1}^2C}{B_2m^2c_0\eta_t}\right.$$
$$\left. + \left(\frac{5\alpha_{t+1}C}{mc_0} + \frac{B_1}{c_0\eta_t m^2} - \frac{B_1^2\beta_{t+1}}{m^3c_0\eta_t} - \frac{B_1}{c_0\eta_{t-1}m^2} + \frac{16B_1\beta_{t+1}^2C}{m^2c_0}\right)\|\mathbf{u}_t - g(\mathbf{w}_t)\|^2\right]$$

By setting $\beta_{t+1} = \frac{256m^2C^2\eta_t^2}{B_1^2}$ (and note that $c_0 = 72C$, $\alpha_{t+1} = 36C\eta_t$), we have:

$$\mathbb{E}\left[\Gamma_{t+1} - \Gamma_t\right] \leq \mathbb{E}\left[-\frac{\eta_t}{2}\|\nabla F(\mathbf{w}_t)\|^2 + \frac{2\alpha_{t+1}^2 C}{\min\{B_1, B_2\}c_0} + \frac{4B_1^2\beta_{t+1}^2 C}{B_2 m^2 c_0 \eta_t}\right]$$
$$\leq \mathbb{E}\left[-\frac{\eta_t}{2}\|\nabla F(\mathbf{w}_t)\|^2 + \frac{36C^2\eta_t^2}{\min\{B_1, B_2\}} + \frac{16^4 m^2 C^4 \eta_t^3}{18 B_2 B_1^2}\right]$$

This means that, by setting $\eta_t = \min\{\sqrt{\min\{B_1, B_2\}}(a+t)^{-1/2}, \left(\frac{B_1\sqrt{B_2}}{m}\right)^{2/3}(a+t)^{-1/3}\}$:

$$\frac{\eta_T}{2}\mathbb{E}\left[\sum_{t=1}^{T}\|\nabla F(\mathbf{w}_t)\|^2\right] \leq \mathbb{E}\left[\Gamma_1 - \Gamma_{T+1}\right] + \frac{36C^2}{\min\{B_1, B_2\}}\mathbb{E}\left[\sum_{t=1}^{T}\eta_t^2\right] + \frac{16^4 m^2 C^4}{18 B_2 B_1^2}\mathbb{E}\left[\sum_{t=1}^{T}\eta_t^3\right]$$
$$\leq \mathbb{E}\left[\Gamma_1\right] + 16^3 C^4 \mathbb{E}\left[\sum_{t=1}^{T}(a+t)^{-1}\right]$$
$$\leq \Delta_F + \frac{2C}{c_0\eta_0} + 16^3 C^4 \ln(1+T)$$

Similar to the proof of Theorem 1 in STORM [Cutkosky and Orabona, 2019], denote $M = \Delta_F + \frac{2C}{c_0\eta_0} + 16^3 C^4 \ln(1+T)$. Using Cauchy-Schwarz inequality, we have:

$$\mathbb{E}\left[\sqrt{\sum_{t=1}^{T}\|\nabla F(\mathbf{w}_t)\|^2}\right]^2 \leq \mathbb{E}\left[1/\eta_T\right]\mathbb{E}\left[\eta_T\sum_{t=1}^{T}\|\nabla F(\mathbf{w}_t)\|^2\right] \leq \mathbb{E}\left[\frac{M}{\eta_T}\right]$$
$$\leq \mathbb{E}\left[M\max\left\{\frac{1}{\sqrt{\min\{B_1, B_2\}}}(a+T)^{1/2}, \left(\frac{m}{B_1\sqrt{B_2}}\right)^{2/3}(a+T)^{1/3}\right\}\right],$$

which indicate that

$$\mathbb{E}\left[\sqrt{\sum_{t=1}^{T}\|\nabla F(\mathbf{w}_t)\|^2}\right]$$
$$\leq \sqrt{M}\max\left\{(\min\{B_1, B_2\})^{-1/4}(a+T)^{1/4}, \left(\frac{m}{B_1\sqrt{B_2}}\right)^{1/3}(a+T)^{1/6}\right\}.$$

Finally, using Cauchy-Schwarz we have $\sum_{t=1}^{T}\|\nabla F(\mathbf{w}_t)\|/T \leq \sqrt{\sum_{t=1}^{T}\|\nabla F(\mathbf{w}_t)\|^2}/\sqrt{T}$ so that:

$$\mathbb{E}\left[\sum_{t=1}^{T}\frac{\|\nabla F(\mathbf{w}_t)\|}{T}\right]$$
$$\leq \max\left\{\sqrt{M}(\min\{B_1, B_2\})^{-1/4}\frac{(a+T)^{1/4}}{\sqrt{T}}, \sqrt{M}\left(\frac{m}{B_1\sqrt{B_2}}\right)^{1/3}\frac{(a+T)^{1/6}}{\sqrt{T}}\right\}$$
$$\leq \max\left\{\sqrt{M}(\min\{B_1, B_2\})^{-1/4}\left(\frac{a^{1/4}}{\sqrt{T}} + \frac{1}{T^{1/4}}\right), \sqrt{M}\left(\frac{m}{B_1\sqrt{B_2}}\right)^{1/3}\left(\frac{a^{1/6}}{\sqrt{T}} + \frac{1}{T^{1/3}}\right)\right\}$$
$$\leq \mathcal{O}\left(\max\left\{\left(\frac{1}{\min\{B_1, B_2\}T}\right)^{1/4}, \left(\frac{m}{B_1\sqrt{B_2}T}\right)^{1/3}\right\}\right),$$

where the last inequality is due to $(a+b)^{1/3} \leq a^{1/3} + b^{1/3}$. So, we can achieve the stationary point with $T = \mathcal{O}\left(\max\left\{\frac{m}{B_1\sqrt{B_2}\epsilon^3}, \frac{1}{\min\{B_1, B_2\}\epsilon^4}\right\}\right)$.

### A.5 Proof of Theorem 2

**Lemma 7** *Denote $\|\mathbf{u}_t - g(\mathbf{w}_t)\|^2 = \sum_{i=1}^{m} \|\mathbf{u}_t^i - g_i(\mathbf{w}_t)\|^2$ and $\|\mathbf{u}_t - \mathbf{u}_{t-1}\|^2 = \sum_{i=1}^{m} \|\mathbf{u}_t^i - \mathbf{u}_{t-1}^i\|^2$.*

$$
\mathbb{E}\left[\|\mathbf{z}_t - \nabla F(\mathbf{w}_t)\|^2\right] \leq 4\mathbb{E}\left[\left\|\mathbf{z}_t - \frac{1}{m}\sum_{i=1}^{m}\nabla g_i(\mathbf{w}_t)\nabla f_i(\mathbf{u}_{t-1}^i)\right\|^2\right]
$$
$$
+ \frac{4C_g^2 L_f^2}{m}\mathbb{E}\left[\|\mathbf{u}_t - \mathbf{u}_{t-1}\|^2\right] + \frac{2C_g^2 L_f^2}{m}\mathbb{E}\left[\|\mathbf{u}_t - g(\mathbf{w}_t)\|^2\right]
$$

**Proof**

$$
\mathbb{E}\left[\|\mathbf{z}_t - \nabla F(\mathbf{w}_t)\|^2\right]
$$
$$
=2\mathbb{E}\left[\left\|\mathbf{z}_t - \frac{1}{m}\sum_{i=1}^{m}\nabla g_i(\mathbf{w}_t)\nabla f_i(\mathbf{u}^i)\right\|^2\right.
$$
$$
+ 2\left\|\frac{1}{m}\sum_{i=1}^{m}\nabla g_i(\mathbf{w}_t)\nabla f_i(\mathbf{u}^i) - \frac{1}{m}\sum_{i=1}^{m}\nabla g_i(\mathbf{w}_t)\nabla f_i(g_i(\mathbf{w}_t))\right\|^2\right]
$$
$$
\leq\mathbb{E}\left[2\left\|\mathbf{z}_t - \frac{1}{m}\sum_{i=1}^{m}\nabla g_i(\mathbf{w}_t)\nabla f_i(\mathbf{u}^i)\right\|^2 + \frac{2}{m}\sum_{i=1}^{m}\left\|\nabla g_i(\mathbf{w}_t)\nabla f_i(\mathbf{u}^i) - \nabla g_i(\mathbf{w}_t)\nabla f_i(g_i(\mathbf{w}_t))\right\|^2\right]
$$
$$
\leq\mathbb{E}\left[2\left\|\mathbf{z}_t - \frac{1}{m}\sum_{i=1}^{m}\nabla g_i(\mathbf{w}_t)\nabla f_i(\mathbf{u}^i)\right\|^2 + \frac{2C_g^2 L_f^2}{m}\sum_{i=1}^{m}\left\|\mathbf{u}_t^i - g_i(\mathbf{w}_t)\right\|^2\right]
$$
$$
\leq\mathbb{E}\left[4\left\|\mathbf{z}_t - \frac{1}{m}\sum_{i=1}^{m}\nabla g_i(\mathbf{w}_t)\nabla f_i(\mathbf{u}_{t-1}^i)\right\|^2 + \frac{4C_g^2 L_f^2}{m}\left\|\mathbf{u}_t - \mathbf{u}_{t-1}\right\|^2 + \frac{2C_g^2 L_f^2}{m}\left\|\mathbf{u}_t - g(\mathbf{w}_t)\right\|^2\right]
$$

**Lemma 8**

$$
\mathbb{E}\left[\left\|\mathbf{z}_t - \frac{1}{m}\sum_{i=1}^{m}\nabla g_i(\mathbf{w}_t)\nabla f_i(\mathbf{u}_{t-1}^i)\right\|^2\right] \leq \mathbb{E}\left[(1-\alpha_t)\left\|\mathbf{z}_{t-1} - \frac{1}{m}\sum_{i=1}^{m}\nabla g_i(\mathbf{w}_{t-1})\nabla f_i(\mathbf{u}_{t-2}^i)\right\|^2\right.
$$
$$
\left. + \frac{2\alpha_t^2\sigma^2}{B_1} + \frac{4C_g^2 L_f^2}{m}\|\mathbf{u}_{t-1} - \mathbf{u}_{t-2}\|^2 + 4C_f^2 L_g^2\|\mathbf{w}_t - \mathbf{w}_{t-1}\|^2\right]
$$

**Proof**

$$
\mathbb{E}\left[\|\mathbf{z}_t - \frac{1}{m}\sum_{i=1}^{m}\nabla g_i(\mathbf{w}_t)\nabla f_i(\mathbf{u}_{t-1}^i)\|^2\right]
$$
$$
=\mathbb{E}\left[\left\|(1-\alpha_t)\left(\mathbf{z}_{t-1} - \frac{1}{m}\sum_{i=1}^{m}\nabla g_i(\mathbf{w}_{t-1})\nabla f_i(\mathbf{u}_{t-2}^i)\right)\right.\right.
$$
$$
+\alpha_t\left(\frac{1}{B_1}\sum_{i\in\mathcal{B}_1^t}\nabla g_i(\mathbf{w}_t;\xi_t^i)\nabla f_i(\mathbf{u}_{t-1}^i) - \frac{1}{m}\sum_{i=1}^{m}\nabla g_i(\mathbf{w}_t)\nabla f_i(\mathbf{u}_{t-1}^i)\right)
$$
$$
+(1-\alpha_t)\left(\frac{1}{B_1}\sum_{i\in\mathcal{B}_1^t}\nabla g_i(\mathbf{w}_t;\xi_t^i)\nabla f_i(\mathbf{u}_{t-1}^i) - \frac{1}{B_1}\sum_{i\in\mathcal{B}_1^t}\nabla g_i(\mathbf{w}_{t-1};\xi_t^i)\nabla f_i(\mathbf{u}_{t-2}^i)\right.
$$
$$
\left.\left.\left. - \frac{1}{m}\sum_{i=1}^{m}\nabla g_i(\mathbf{w}_t)\nabla f_i(\mathbf{u}_{t-1}^i) + \frac{1}{m}\sum_{i=1}^{m}\nabla g_i(\mathbf{w}_{t-1})\nabla f_i(\mathbf{u}_{t-2}^i)\right)\right\|^2\right]
$$

We assume that $\mathbb{E}\left[\|\frac{1}{B_1}\sum_{i\in\mathcal{B}_1^t}\nabla g_i(\mathbf{w}_t;\xi_t^i)\nabla f_i(\mathbf{u}_{t-1}^i) - \frac{1}{m}\sum_{i=1}^m\nabla g_i(\mathbf{w}_t)\nabla f_i(\mathbf{u}_{t-1}^i)\|^2\right] \leq \frac{\sigma^2}{B_1}$.
Due to the fact that the expectation over the last two terms equals zero, we have:

$$
\mathbb{E}\left[\left\|\mathbf{z}_t - \frac{1}{m}\sum_{i=1}^m\nabla g_i(\mathbf{w}_t)\nabla f_i(\mathbf{u}_{t-1}^i)\right\|^2\right]
$$

$$
\leq \mathbb{E}\left[(1-\alpha_t)^2\left\|\mathbf{z}_{t-1} - \frac{1}{m}\sum_{i=1}^m\nabla g_i(\mathbf{w}_{t-1})\nabla f_i(\mathbf{u}_{t-2}^i)\right\|^2 + \frac{2\alpha_t^2\sigma^2}{B_1}\right.
$$

$$
\left. + 2(1-\alpha_t)^2\frac{1}{B_1}\sum_{i\in\mathcal{B}_1^t}\left\|\nabla g_i(\mathbf{w}_t;\xi_t^i)\nabla f_i(\mathbf{u}_{t-1}^i) - \nabla g_i(\mathbf{w}_{t-1};\xi_t^i)\nabla f_i(\mathbf{u}_{t-2}^i)\right\|^2\right]
$$

$$
\leq \mathbb{E}\left[(1-\alpha_t)\left\|\mathbf{z}_{t-1} - \frac{1}{m}\sum_{i=1}^m\nabla g_i(\mathbf{w}_{t-1})\nabla f_i(\mathbf{u}_{t-2}^i)\right\|^2 + \frac{2\alpha_t^2\sigma^2}{B_1}\right.
$$

$$
+ 4(1-\alpha_t)^2\frac{1}{B_1}\sum_{i\in\mathcal{B}_1^t}\left\|\nabla g_i(\mathbf{w}_t;\xi_t^i)\left(\nabla f_i(\mathbf{u}_{t-1}^i) - \nabla f_i(\mathbf{u}_{t-2}^i)\right)\right\|^2
$$

$$
\left. + 4(1-\alpha_t)^2\frac{1}{B_1}\sum_{i\in\mathcal{B}_1^t}\left\|\nabla f_i(\mathbf{u}_{t-2}^i)\left(\nabla g_i(\mathbf{w}_t;\xi_t^i) - \nabla g_i(\mathbf{w}_{t-1};\xi_t^i)\right)\right\|^2\right]
$$

$$
\leq \mathbb{E}\left[(1-\alpha_t)\left\|\mathbf{z}_{t-1} - \frac{1}{m}\sum_{i=1}^m\nabla g_i(\mathbf{w}_{t-1})\nabla f_i(\mathbf{u}_{t-2}^i)\right\|^2 + \frac{2\alpha_t^2\sigma^2}{B_1} + \frac{4C_g^2L_f^2}{m}\|\mathbf{u}_{t-1}-\mathbf{u}_{t-2}\|^2\right.
$$

$$
\left. + 4C_f^2L_g^2\|\mathbf{w}_t-\mathbf{w}_{t-1}\|^2\right]
$$

**Lemma 9** *Suppose that $\beta \leq \frac{1}{32C}$ and $B_1\beta_{t+1} \leq m\alpha_{t+1}$. Then, we have:*

$$
\mathbb{E}\left[\frac{1}{m}\|\mathbf{u}_{t+1} - g(\mathbf{w}_{t+1})\|^2 + \left\|\mathbf{z}_{t+1} - \frac{1}{m}\sum_{i=1}^m\nabla g_i(\mathbf{w}_{t+1})\nabla f_i(\mathbf{u}_t^i)\right\|^2\right]
$$

$$
\leq (1-\frac{B_1\beta_{t+1}}{m})\frac{1}{m}\mathbb{E}\left[\|\mathbf{u}_t - g(\mathbf{w}_t)\|^2\right] + \frac{8mC_g^2}{B_1}\|\mathbf{w}_{t+1}-\mathbf{w}_t\|^2 + \frac{2B_1\beta_{t+1}^2\sigma^2}{B_2m}
$$

$$
+ (1-\alpha_{t+1})\mathbb{E}\left[\left\|\mathbf{z}_t - \frac{1}{m}\sum_{i=1}^m\nabla g_i(\mathbf{w}_t)\nabla f_i(\mathbf{u}_{t-1}^i)\right\|^2\right] + \frac{2\alpha_{t+1}^2\sigma^2}{B_1}
$$

$$
+ \frac{4C_g^2L_f^2}{m}\mathbb{E}\left[\|\mathbf{u}_t-\mathbf{u}_{t-1}\|^2\right] + 4C_f^2L_g^2\mathbb{E}\left[\|\mathbf{w}_{t+1}-\mathbf{w}_t\|^2\right]
$$

$$
\leq (1-\frac{B_1\beta_{t+1}}{m})\mathbb{E}\left[\frac{1}{m}\|\mathbf{u}_t - g(\mathbf{w}_t)\|^2 + \left\|\mathbf{z}_t - \frac{1}{m}\sum_{i=1}^m\nabla g_i(\mathbf{w}_t)\nabla f_i(\mathbf{u}_{t-1}^i)\right\|^2\right]
$$

$$
+ \frac{12mC}{B_1}\|\mathbf{w}_{t+1}-\mathbf{w}_t\|^2 + \frac{2B_1\beta_{t+1}^2\sigma^2}{B_2m} + \frac{2\alpha_{t+1}^2\sigma^2}{B_1} + \frac{4C_g^2L_f^2}{m}\mathbb{E}\left[\|\mathbf{u}_t-\mathbf{u}_{t-1}\|^2\right]
$$

$$
\leq (1-\frac{B_1\beta_{t+1}}{2m})\mathbb{E}\left[\frac{1}{m}\|\mathbf{u}_t - g(\mathbf{w}_t)\|^2 + \left\|\mathbf{z}_t - \frac{1}{m}\sum_{i=1}^m\nabla g_i(\mathbf{w}_t)\nabla f_i(\mathbf{u}_{t-1}^i)\right\|^2\right]
$$

$$
+ \frac{48mC}{B_1}\|\mathbf{w}_{t+1}-\mathbf{w}_t\|^2 + \frac{10B_1\beta_{t+1}^2C}{B_2m} + \frac{2\alpha_{t+1}^2C}{B_1}
$$

**The rest proof of Theorem 2** Set $\eta_t \leq \frac{B_1}{mc_0}$. Denote $\Gamma_t = F(\mathbf{w}_t) + \frac{B_1}{c_0 \eta_t m} \Delta_t$, where $\Delta_t = \frac{1}{m} \|\mathbf{u}_t - g(\mathbf{w}_t)\|^2 + \|\mathbf{z}_t - \frac{1}{m} \sum_{i=1}^m \nabla g_i(\mathbf{w}_t) \nabla f_i(\mathbf{u}_{t-1}^i)\|^2$. We have:

$$\mathbb{E}\left[\Gamma_{t+1} - \Gamma_t\right]$$

$$=\mathbb{E}\left[F(\mathbf{w}_{t+1}) - F(\mathbf{w}_t) + \frac{B_1}{c_0 \eta_t m} \Delta_{t+1} - \frac{B_1}{c_0 \eta_{t-1} m} \Delta_t\right]$$

$$\leq \mathbb{E}\left[-\frac{\eta_t}{2} \|\nabla F(\mathbf{w}_t)\|^2 + \frac{\eta_t}{2} \|\mathbf{z}_t - \nabla F(\mathbf{w}_t)\|^2 - \frac{\eta_t}{4} \|\mathbf{z}_t\|^2\right.$$

$$\left. + \left(\frac{B_1}{c_0 \eta_t m} - \frac{B_1^2 \beta_{t+1}}{2m^2 c_0 \eta_t} - \frac{B_1}{c_0 \eta_{t-1} m}\right) \Delta_t + \frac{48C}{c_0 \eta_t} \|\mathbf{w}_{t+1} - \mathbf{w}_t\|^2 + \frac{10 B_1^2 \beta_{t+1}^2 C}{B_2 m^2 c_0 \eta_t} + \frac{2 \alpha_{t+1}^2 C}{m c_0 \eta_t}\right]$$

$$\leq \mathbb{E}\left[-\frac{\eta_t}{2} \|\nabla F(\mathbf{w}_t)\|^2 - \frac{\eta_t}{4} \|\mathbf{z}_t\|^2 + \frac{66C}{c_0 \eta_t} \|\mathbf{w}_{t+1} - \mathbf{w}_t\|^2 + \frac{14 B_1^2 \beta_{t+1}^2 C}{B_2 m^2 c_0 \eta_t} + \frac{2 \alpha_{t+1}^2 C}{m c_0 \eta_t}\right.$$

$$\left. + \left(2C\eta_t + \frac{B_1}{c_0 \eta_t m} - \frac{B_1^2 \beta_{t+1}}{2m^2 c_0 \eta_t} - \frac{B_1}{c_0 \eta_{t-1} m}\right) \Delta_t\right]$$

By setting $264C = c_0$, $\eta_t^2 = \frac{32 B_1^2 \beta_{t+1}}{m^2 c_0^2}$, $\alpha_{t+1} = \frac{B_1 \beta_{t+1}}{m}$ and $B_2 \leq m$, we have:

$$\mathbb{E}\left[\Gamma_{t+1} - \Gamma_t\right] \leq \mathbb{E}\left[-\frac{\eta_t}{2} \|\nabla F(\mathbf{w}_t)\|^2 + \frac{14 B_1^2 \beta_{t+1}^2 C}{B_2 m^2 c_0 \eta_t} + \frac{2 \alpha_{t+1}^2 C}{m c_0 \eta_t}\right]$$

$$\leq \mathbb{E}\left[-\frac{\eta_t}{2} \|\nabla F(\mathbf{w}_t)\|^2 + \frac{m^2 \eta_t^3 c_0^4}{512 B_2 B_1^2}\right]$$

This means that, by setting $\eta_t = \left(\frac{B_1 \sqrt{B_2}}{m}\right)^{\frac{2}{3}} (a+t)^{-\frac{1}{3}}$

$$\frac{\eta_T}{2} \mathbb{E}\left[\sum_{t=1}^T \|\nabla F(\mathbf{w}_t)\|^2\right] \leq \mathbb{E}\left[\Gamma_1 - \Gamma_{T+1} + \frac{m^2 c_0^4}{512 B_2 B_1^2} \sum_{t=1}^T \eta_t^3\right]$$

$$\leq \mathbb{E}\left[\Gamma_1 + \frac{c_0^4}{512} \sum_{t=1}^T (a+t)^{-1}\right]$$

$$\leq \Delta_F + \frac{1}{8\eta_0} + \frac{c_0^4}{512} \ln(1+T)$$

Denote $M = \Delta_F + \frac{1}{8\eta_0} + \frac{c_0^4}{512} \ln(1+T)$. Using Cauchy-Schwarz inequality, we have:

$$\mathbb{E}\left[\sqrt{\sum_{t=1}^T \|\nabla F(\mathbf{w}_t)\|^2}\right]^2 \leq \mathbb{E}[1/\eta_T] \mathbb{E}\left[\eta_T \sum_{t=1}^T \|\nabla F(\mathbf{w}_t)\|^2\right] \leq \mathbb{E}\left[\frac{M}{\eta_T}\right]$$

$$\leq \mathbb{E}\left[M \left(\frac{m}{B_1 \sqrt{B_2}}\right)^{2/3} (a+T)^{1/3}\right],$$

which indicate that

$$\mathbb{E}\left[\sqrt{\sum_{t=1}^T \|\nabla F(\mathbf{w}_t)\|^2}\right] \leq \sqrt{M} \left(\frac{m}{B_1 \sqrt{B_2}}\right)^{1/3} (a+T)^{1/6}.$$

Finally, using Cauchy-Schwarz we have $\sum_{t=1}^T \|\nabla F(\mathbf{w}_t)\|/T \leq \sqrt{\sum_{t=1}^T \|\nabla F(\mathbf{w}_t)\|^2}/\sqrt{T}$ so that:

$$\mathbb{E}\left[\sum_{t=1}^T \frac{\|\nabla F(\mathbf{w}_t)\|}{T}\right] \leq \frac{\sqrt{M}(a+T)^{1/6}}{\sqrt{T}} \left(\frac{m}{B_1 \sqrt{B_2}}\right)^{1/3} \leq \mathcal{O}\left(\frac{a^{1/6}\sqrt{M}}{\sqrt{T}} + \left(\frac{m}{B_1 \sqrt{B_2} T}\right)^{1/3}\right)$$

$$= \mathcal{O}\left(\left(\frac{m}{T B_1 \sqrt{B_2}}\right)^{1/3}\right),$$

where the last inequality is due to $(a+b)^{1/3} \leq a^{1/3} + b^{1/3}$. So, we can achieve the stationary point with $T = \mathcal{O}\left(m/B_1 \sqrt{B_2}\epsilon^3\right)$.

---

**Algorithm 3** Stage-wise MSVR method

---

**Input:** initial points $(\mathbf{w}_0, \mathbf{u}_0, \mathbf{z}_0)$
**for** stage $s = 1$ **to** $S$ **do**
    $\mathbf{w}_s, \mathbf{u}_s, \mathbf{z}_s =$ MSVR (with $T_s$, $\alpha_s$, $\beta_s$, $\eta_s$ and $(\mathbf{w}_{s-1}, \mathbf{u}_{s-1}, \mathbf{z}_{s-1})$)
**end for**
Return $\mathbf{w}_S$

---

## A.6 Proof of Theorem 3

We would show that the complexity can be further improved if the objective function satisfies the Polyak-Łojasiewicz (PL) condition or convexity. To achieve this, we utilize the previous analysis and use a stage-wise version method [Yuan et al., 2019b]. In the new algorithm, we decrease $\alpha_s$ and $\beta_s$ after each stage and increase the number of iterations $T_s$. At the end of each stage, we save the output and use it to restart the next stage. With these modifications, we can obtain a better convergence guarantee under the PL condition or convexity. The new method is summarized in Algorithm 3, named Stage-wise MSVR. Next, we will show the proof for optimal MSVR-v2 with Stage-wise version, and the proof for MSVR-v1 is nearly the same as the MSVR-v2.

Note that in below the numerical subscripts denote the stage index $\{1, \ldots, S\}$. Denote $\Delta_s = \left\| \mathbf{z}_s - \frac{1}{m} \sum_{i=1}^{m} \nabla g_i(\mathbf{w}_s) \nabla f_i(\mathbf{u}_{s-1}^i) \right\|^2 + \frac{1}{m} \left\| \mathbf{u}_s - g(\mathbf{w}_s) \right\|^2$. Let's consider the first stage, $\Delta_1 \leq 2C = \mu \epsilon_1$ and $F(\mathbf{w}_1) - F_* \leq \epsilon_1$, where $\epsilon_1 = \max\{\frac{2C}{\mu}, \Delta_F\}$. Starting form the second stage, we would prove by induction.

Suppose at stage $s - 1$, we have $\Delta_{s-1} \leq \mu \epsilon_{s-1}$ and $F(\mathbf{w}_{s-1}) - F_* \leq \epsilon_{s-1}$. Then at $s$ stage, by setting $264C = c_0$, $\eta_s^2 = \frac{32 B_1^2 \beta_s}{m^2 c_0^2}$, $\alpha_s = \frac{B_1 \beta_s}{m}$ and $B_2 \leq m$, we have:

$$\mathbb{E}\left[\Gamma_{t+1} - \Gamma_t\right] \leq \mathbb{E}\left[-\frac{\eta_s}{2}\|\nabla F(\mathbf{w}_t)\|^2 + \frac{m^2 \eta_s^3 c_0^4}{512 B_2 B_1^2}\right]$$

This means that by setting $T_s = \max\left\{\frac{mc_0^2}{B_1 \mu \sqrt{B_2 \mu \epsilon_s}}, \frac{mc_0^4}{B_1 B_2 \mu \epsilon_s}\right\}$, $\eta_s = \frac{8 B_1 \sqrt{B_2 \mu \epsilon_s}}{mc_0^2}$, we have:

$$
\frac{1}{T}\mathbb{E}\left[\sum_{t=1}^{T} \|\nabla F(\mathbf{w}_t)\|^2\right]
$$
$$
\leq \mathbb{E}\left[\frac{2(\Gamma_1 - \Gamma_{T+1})}{\eta_s T} + \frac{m^2 c_0^4 \eta_s^2}{256 B_2 B_1^2}\right]
$$
$$
\leq \mathbb{E}\left[\frac{2(F(\mathbf{w}_{s-1}) - F_*)}{\eta_s T} + \frac{2 B_1 \Delta_{s-1}}{c_0 \eta_s^2 T m} + \frac{m^2 c_0^4 \eta_s^2}{256 B_2 B_1^2}\right]
$$
$$
\leq 2 \mu_s \epsilon_s
$$

Due to the PL condition, we have:

$$F(\mathbf{w}_s) - F_* \leq \frac{1}{2\mu T}\mathbb{E}\left[\sum_{t=1}^{T} \|\nabla F(\mathbf{w}_t)\|^2\right] \leq \epsilon_s$$

On the other hand, by setting $\beta_s = \frac{B_2 \mu \epsilon_s}{80C}$ and $\alpha_s = \frac{B_1 \beta_s}{m}$, we have:

$$
\Delta_s \leq \frac{2m}{B_1 \beta_s T}\Delta_{s-1} + \frac{96 m^2 C}{B_1^2 \beta_s T}\sum_{t=1}^{T}\|\mathbf{w}_{t+1} - \mathbf{w}_t\|^2 + \frac{20\beta_s C}{B_2} + \frac{4m\alpha_s^2 C}{B_1^2 \beta}
$$
$$
\leq \frac{2m\mu\epsilon_{s-1}}{B_1 \beta_s T} + \frac{96 m^2 \eta_s^2 C}{B_1^2 \beta_s T}\sum_{t=1}^{T}\|\mathbf{z}_t\|^2 + \frac{20\beta_s C}{B_2} + \frac{4m\alpha_s^2 C}{B_1^2 \beta}
$$
$$
\leq \mu\epsilon_s
$$

So, we proved that $F(\mathbf{w}_s) - F_* \leq \epsilon_s$. That is to say, $F(\mathbf{w}_s) - F_* \leq \epsilon$ when $S = \log_2\left(\frac{2\epsilon_1}{\epsilon}\right)$, and the iteration complexity is computed as:

$$T_1 + \sum_{s=2}^{S} T_s \overset{\mu \geq \epsilon}{=} \mathcal{O}\left(\sum_{s=2}^{S} \frac{m}{B_1\sqrt{B_2}\mu\epsilon_s}\right)$$

$$\leq \mathcal{O}\left(\frac{m}{B_1\sqrt{B_2}\mu\epsilon}\right)$$

When $F(\mathbf{w})$ is convex, we define $\hat{F}(\mathbf{w}) = F(\mathbf{w}) + \frac{\mu}{2}\|\mathbf{w}\|^2$. We know that $\hat{F}(\mathbf{w})$ is $\mu$-strongly convex, which implies $\mu$-PL condition. We have proved: for any $\delta > 0$, there exist $T = \mathcal{O}\left(\frac{m}{\mu\delta}\right)$ such that $\hat{F}(\mathbf{w}_T) - \hat{F}_* \leq \delta$. It indicates that $F(\mathbf{w}_T) - F_* \leq \delta + \frac{\mu}{2}\|\mathbf{w}_*\|^2 - \frac{\mu}{2}\|\mathbf{w}_T\|^2 \leq \delta + \frac{\mu}{2}D$. For any $\epsilon > 0$, if we choose $\mu = \frac{\epsilon}{D}$ and $\delta = \frac{\epsilon}{2}$, we get $F(\mathbf{w}_T) - F_* \leq \epsilon$, for some $T = \mathcal{O}\left(\frac{m}{\epsilon^2}\right)$.

## A.7 Proof of Theorem 4

**Lemma 10** *If $\beta \leq \frac{1}{2}$ and $\beta I \leq \frac{m}{B_1}$, we have:*

$$\mathbb{E}\left[\sum_{t=1}^{T}\|\mathbf{u}_{t+1} - \mathbf{u}_t\|^2\right] \leq \frac{4B_1\beta^2}{m}\mathbb{E}\left[\sum_{t=1}^{T}\|\mathbf{u}_t - g(\mathbf{w}_t)\|^2\right] + \frac{11m^2C_g^2}{B_1}\sum_{t=1}^{T}\|\mathbf{w}_{t+1} - \mathbf{w}_t\|^2$$

**Proof** *Following the analysis of Lemma 6, we have:*

$$\mathbb{E}\left[\|\mathbf{u}_{t+1} - \mathbf{u}_t\|^2\right] \leq \frac{2B_1\beta^2}{m}\sum_{i=1}^{m}\left(\mathbb{E}\left[\|\widehat{g}_i(\mathbf{w}_{t+1};\xi_{t+1}^i) - g_i(\mathbf{w}_{t+1})\|^2\right] + \mathbb{E}\left[\|g_i(\mathbf{w}_{t+1}) - \mathbf{u}_t^i\|^2\right]\right)$$
$$+ \frac{8m^2C_g^2}{B_1}\|\mathbf{w}_{t+1} - \mathbf{w}_t\|^2$$

*So, we have:*

$$\mathbb{E}\left[\|\mathbf{u}_{t+1} - \mathbf{u}_t\|^2\right] \leq 2B_1\beta^2 C_g^2\|\mathbf{w}_{t+1} - \mathbf{w}_{\tau+1}\|^2 + \frac{2B_1\beta^2}{m}\mathbb{E}\left[\|g(\mathbf{w}_{t+1}) - \mathbf{u}_t\|^2\right]$$
$$+ \frac{8m^2C_g^2}{B_1}\|\mathbf{w}_{t+1} - \mathbf{w}_t\|^2$$

*So, with $\beta^2 I^2 \leq m^2/B_1^2$, we have:*

$$\mathbb{E}\left[\sum_{t=1}^{T}\|\mathbf{u}_{t+1} - \mathbf{u}_t\|^2\right] \leq \frac{2B_1\beta^2}{m}\mathbb{E}\left[\sum_{t=1}^{T}\|g(\mathbf{w}_{t+1}) - \mathbf{u}_t\|^2\right]$$
$$+ 2B_1\beta^2 C_g^2\sum_{t=1}^{T}\|\mathbf{w}_{t+1} - \mathbf{w}_{\tau+1}\|^2 + \frac{8m^2C_g^2}{B_1}\sum_{t=1}^{T}\|\mathbf{w}_{t+1} - \mathbf{w}_t\|^2$$
$$\leq \frac{4B_1\beta^2}{m}\mathbb{E}\left[\sum_{t=1}^{T}\|g(\mathbf{w}_t) - \mathbf{u}_t\|^2\right]$$
$$+ 2B_1\beta^2 C_g^2 I^2\sum_{t=1}^{T}\|\mathbf{w}_{t+1} - \mathbf{w}_t\|^2 + \frac{9m^2C_g^2}{B_1}\sum_{t=1}^{T}\|\mathbf{w}_{t+1} - \mathbf{w}_t\|^2$$
$$\leq \frac{4B_1\beta^2}{m}\mathbb{E}\left[\sum_{t=1}^{T}\|g(\mathbf{w}_t) - \mathbf{u}_t\|^2\right] + \frac{11m^2C_g^2}{B_1}\sum_{t=1}^{T}\|\mathbf{w}_{t+1} - \mathbf{w}_t\|^2$$

We can also replace Lemma 8 with following lemma.

**Lemma 11** *With $\alpha I \leq 1$, we have:*

$$\mathbb{E}\left[\sum_{t=1}^{T}\left\|\mathbf{z}_t - \frac{1}{m}\sum_{i=1}^{m}\nabla g_i(\mathbf{w}_t)\nabla f_i(\mathbf{u}_{t-1}^i)\right\|^2\right] \leq \frac{1}{\alpha}\left\|\mathbf{z}_1 - \frac{1}{m}\sum_{i=1}^{m}\nabla g_i(\mathbf{w}_1)\nabla f_i(\mathbf{u}_0^i)\right\|^2$$
$$+ \frac{8C_g^2 L_f^2}{m\alpha}\mathbb{E}\left[\sum_{t=1}^{T}\|\mathbf{u}_t - \mathbf{u}_{t-1}\|^2\right] + \frac{8C_f^2 L_g^2}{\alpha}\mathbb{E}\left[\sum_{t=1}^{T}\|\mathbf{w}_{t+1} - \mathbf{w}_t\|^2\right]$$

**Proof** *First, since $\mathbf{h}_t$ is an unbiased estimation of $\frac{1}{m}\sum_{i=1}^m \nabla g_i(\mathbf{w}_t)\nabla f_i(\mathbf{u}_{t-1}^i)$, we have:*

$$
\mathbb{E}\left[\left\|\mathbf{h}_t - \frac{1}{m}\sum_{i=1}^m \nabla g_i(\mathbf{w}_t)\nabla f_i(\mathbf{u}_{t-1}^i)\right\|^2\right]
$$

$$
=\mathbb{E}\left[\left\|\frac{1}{B_1}\sum_{i\in\mathcal{B}_1^t}\nabla f_i(\mathbf{u}_{t-1}^i)\nabla g_i(\mathbf{w}_t;\xi_t^i) - \frac{1}{B_1}\sum_{i\in\mathcal{B}_1^t}\nabla f_i(\mathbf{u}_{\tau-1}^i)\nabla g_i(\mathbf{w}_\tau;\xi_t^i)\right.\right.
$$

$$
\left.\left.+\frac{1}{m}\sum_{i=1}^m \nabla f_i(\mathbf{u}_{\tau-1}^i)\nabla g_i(\mathbf{w}_\tau;\xi_t^i) - \frac{1}{m}\sum_{i=1}^m \nabla g_i(\mathbf{w}_t)\nabla f_i(\mathbf{u}_{t-1}^i)\right\|^2\right]
$$

$$
\leq\mathbb{E}\left[\left\|\frac{1}{B_1}\sum_{i\in\mathcal{B}_1^t}\nabla f_i(\mathbf{u}_{t-1}^i)\nabla g_i(\mathbf{w}_t;\xi_t^i) - \frac{1}{B_1}\sum_{i\in\mathcal{B}_1^t}\nabla f_i(\mathbf{u}_{\tau-1}^i)\nabla g_i(\mathbf{w}_\tau;\xi_t^i)\right\|^2\right]
$$

$$
\leq\mathbb{E}\left[\frac{1}{B_1}\sum_{i\in\mathcal{B}_1^t}\left\|\nabla f_i(\mathbf{u}_{t-1}^i)\nabla g_i(\mathbf{w}_t;\xi_t^i) - \nabla f_i(\mathbf{u}_{\tau-1}^i)\nabla g_i(\mathbf{w}_\tau;\xi_t^i)\right\|^2\right]
$$

$$
=2C_f^2 L_g^2\|\mathbf{w}_t - \mathbf{w}_\tau\|^2 + \frac{2C_g^2 L_f^2}{m}\|\mathbf{u}_{t-1} - \mathbf{u}_{\tau-1}\|^2
$$

*Next, we have:*

$$
\mathbb{E}\left[\|\mathbf{z}_t - \frac{1}{m}\sum_{i=1}^m \nabla g_i(\mathbf{w}_t)\nabla f_i(\mathbf{u}_{t-1}^i)\|^2\right]
$$

$$
=\mathbb{E}\left[\left\|(1-\alpha)\left(\mathbf{z}_{t-1} - \frac{1}{m}\sum_{i=1}^m \nabla g_i(\mathbf{w}_{t-1})\nabla f_i(\mathbf{u}_{t-2}^i)\right) + \alpha\left(\mathbf{h}_t - \frac{1}{m}\sum_{i=1}^m \nabla g_i(\mathbf{w}_t)\nabla f_i(\mathbf{u}_{t-1}^i)\right)\right.\right.
$$

$$
+(1-\alpha)\left(\frac{1}{B_1}\sum_{i\in\mathcal{B}_1^t}\nabla g_i(\mathbf{w}_t;\xi_t^i)\nabla f_i(\mathbf{u}_{t-1}^i) - \frac{1}{B_1}\sum_{i\in\mathcal{B}_1^t}\nabla g_i(\mathbf{w}_{t-1};\xi_t^i)\nabla f_i(\mathbf{u}_{t-2}^i)\right.
$$

$$
\left.\left.-\frac{1}{m}\sum_{i=1}^m \nabla g_i(\mathbf{w}_t)\nabla f_i(\mathbf{u}_{t-1}^i) + \frac{1}{m}\sum_{i=1}^m \nabla g_i(\mathbf{w}_{t-1})\nabla f_i(\mathbf{u}_{t-2}^i)\right)\right\|^2\right]
$$

$$
\leq\mathbb{E}\left[(1-\alpha)^2\left\|\mathbf{z}_{t-1} - \frac{1}{m}\sum_{i=1}^m \nabla g_i(\mathbf{w}_{t-1})\nabla f_i(\mathbf{u}_{t-2}^i)\right\|^2 + 4\alpha^2 C_f^2 L_g^2\|\mathbf{w}_t - \mathbf{w}_\tau\|^2\right.
$$

$$
+\frac{4\alpha^2 C_g^2 L_f^2}{m}\|\mathbf{u}_{t-1} - \mathbf{u}_{\tau-1}\|^2
$$

$$
\left.+2(1-\alpha)^2\frac{1}{B_1}\sum_{i\in\mathcal{B}_1^t}\left\|\nabla g_i(\mathbf{w}_t;\xi_t^i)\nabla f_i(\mathbf{u}_{t-1}^i) - \nabla g_i(\mathbf{w}_{t-1};\xi_t^i)\nabla f_i(\mathbf{u}_{t-2}^i)\right\|^2\right]
$$

$$
\mathbb{E}\left[\leq(1-\alpha)\|\mathbf{z}_{t-1} - \frac{1}{m}\sum_{i=1}^m \nabla g_i(\mathbf{w}_{t-1})\nabla f_i(\mathbf{u}_{t-2}^i)\|^2 + 4\alpha^2 C_f^2 L_g^2\|\mathbf{w}_t - \mathbf{w}_\tau\|^2\right.
$$

$$
\left.+\frac{4\alpha^2 C_g^2 L_f^2}{m}\|\mathbf{u}_{t-1} - \mathbf{u}_{\tau-1}\|^2 + \frac{4C_g^2 L_f^2}{m}\|\mathbf{u}_{t-1} - \mathbf{u}_{t-2}\|^2 + 4C_f^2 L_g^2\|\mathbf{w}_t - \mathbf{w}_{t-1}\|^2\right]
$$

*The first inequality is due to the fact that the last two terms equal zero in expectation.*

*Summing up, we have:*

$$\sum_{t=1}^{T} \left\| \mathbf{z}_t - \frac{1}{m} \sum_{i=1}^{m} \nabla g_i(\mathbf{w}_t) \nabla f_i(\mathbf{u}_{t-1}^i) \right\|^2$$

$$\leq \frac{1}{\alpha} \left\| \mathbf{z}_1 - \frac{1}{m} \sum_{i=1}^{m} \nabla g_i(\mathbf{w}_1) \nabla f_i(\mathbf{u}_0^i) \right\|^2 + 4\alpha C_f^2 L_g^2 \sum_{t=1}^{T} \|\mathbf{w}_t - \mathbf{w}_\tau\|^2$$

$$+ \frac{4\alpha C_g^2 L_f^2}{m} \sum_{t=1}^{T} \|\mathbf{u}_{t-1} - \mathbf{u}_{\tau-1}\|^2 + \frac{4C_g^2 L_f^2}{m\alpha} \sum_{t=1}^{T} \|\mathbf{u}_t - \mathbf{u}_{t-1}\|^2 + \frac{4C_f^2 L_g^2}{\alpha} \sum_{t=1}^{T} \|\mathbf{w}_{t+1} - \mathbf{w}_t\|^2$$

$$\leq \frac{1}{\alpha} \left\| \mathbf{z}_1 - \frac{1}{m} \sum_{i=1}^{m} \nabla g_i(\mathbf{w}_1) \nabla f_i(\mathbf{u}_0^i) \right\|^2 + 4\alpha C_f^2 L_g^2 I^2 \sum_{t=1}^{T} \|\mathbf{w}_{t+1} - \mathbf{w}_t\|^2$$

$$+ \frac{4\alpha C_g^2 L_f^2 I^2}{m} \sum_{t=1}^{T} \|\mathbf{u}_t - \mathbf{u}_{t-1}\|^2 + \frac{4C_g^2 L_f^2}{m\alpha} \sum_{t=1}^{T} \|\mathbf{u}_t - \mathbf{u}_{t-1}\|^2 + \frac{4C_f^2 L_g^2}{\alpha} \sum_{t=1}^{T} \|\mathbf{w}_{t+1} - \mathbf{w}_t\|^2$$

$$\leq \frac{1}{\alpha} \left\| \mathbf{z}_1 - \frac{1}{m} \sum_{i=1}^{m} \nabla g_i(\mathbf{w}_1) \nabla f_i(\mathbf{u}_0^i) \right\|^2 + \frac{8C_g^2 L_f^2}{m\alpha} \sum_{t=1}^{T} \|\mathbf{u}_t - \mathbf{u}_{t-1}\|^2 + \frac{8C_f^2 L_g^2}{\alpha} \sum_{t=1}^{T} \|\mathbf{w}_{t+1} - \mathbf{w}_t\|^2$$

*The last inequality is due to $\alpha I \leq 1$.*

**The rest proof of Theorem 4**    According to Lemma 7, we have:

$$\sum_{t=1}^{T} \|\mathbf{z}_t - \nabla F(\mathbf{w}_t)\|^2 \leq 4 \sum_{t=1}^{T} \left\| \mathbf{z}_t - \frac{1}{m} \sum_{i=1}^{m} \nabla g_i(\mathbf{w}_t) \nabla f_i(\mathbf{u}_{t-1}^i) \right\|^2$$

$$+ \frac{4C_g^2 L_f^2}{m} \sum_{t=1}^{T} \|\mathbf{u}_t - \mathbf{u}_{t-1}\|^2 + \frac{2C_g^2 L_f^2}{m} \sum_{t=1}^{T} \|\mathbf{u}_t - g(\mathbf{w}_t)\|^2$$

We use Lemma 11 to replace $\sum_{t=1}^{T} \left\| \mathbf{z}_t - \frac{1}{m} \sum_{i=1}^{m} \nabla g_i(\mathbf{w}_t) \nabla f_i(\mathbf{u}_{t-1}^i) \right\|^2$:

$$\mathbb{E} \left[ \sum_{t=1}^{T} \|\mathbf{z}_t - \nabla F(\mathbf{w}_t)\|^2 \right]$$

$$\leq \frac{4}{\alpha} \left\| \mathbf{z}_1 - \frac{1}{m} \sum_{i=1}^{m} \nabla g_i(\mathbf{w}_1) \nabla f_i(\mathbf{u}_0^i) \right\|^2 + \frac{32C_g^2 L_f^2}{m\alpha} \sum_{t=1}^{T} \|\mathbf{u}_t - \mathbf{u}_{t-1}\|^2$$

$$+ \frac{32C_f^2 L_g^2}{\alpha} \sum_{t=1}^{T} \|\mathbf{w}_{t+1} - \mathbf{w}_t\|^2 + \frac{4C_g^2 L_f^2}{m} \sum_{t=1}^{T} \|\mathbf{u}_t - \mathbf{u}_{t-1}\|^2 + \frac{2C_g^2 L_f^2}{m} \sum_{t=1}^{T} \|\mathbf{u}_t - g(\mathbf{w}_t)\|^2$$

$$\leq \frac{4}{\alpha} \left\| \mathbf{z}_1 - \frac{1}{m} \sum_{i=1}^{m} \nabla g_i(\mathbf{w}_1) \nabla f_i(\mathbf{u}_0^i) \right\|^2 + \frac{36C_g^2 L_f^2}{m\alpha} \mathbb{E} \left[ \sum_{t=1}^{T} \|\mathbf{u}_t - \mathbf{u}_{t-1}\|^2 \right]$$

$$+ \frac{32C_f^2 L_g^2}{\alpha} \mathbb{E} \left[ \sum_{t=1}^{T} \|\mathbf{w}_{t+1} - \mathbf{w}_t\|^2 \right] + \frac{2C_g^2 L_f^2}{m} \mathbb{E} \left[ \sum_{t=1}^{T} \|\mathbf{u}_t - g(\mathbf{w}_t)\|^2 \right]$$

Set $\beta B_1 \leq m\alpha$. We use Lemma 10 to replace $\mathbb{E}\left[\sum_{t=1}^T \|\mathbf{u}_t - \mathbf{u}_{t-1}\|^2\right]$ (set $\mathbf{u}_0 = \mathbf{u}_1$):

$$\mathbb{E}\left[\sum_{t=1}^T \|\mathbf{z}_t - \nabla F(\mathbf{w}_t)\|^2\right]$$

$$\leq \frac{4}{\alpha}\left\|\mathbf{z}_1 - \frac{1}{m}\sum_{i=1}^m \nabla g_i(\mathbf{w}_1)\nabla f_i(\mathbf{u}_0^i)\right\|^2 + \frac{144(C_g^2 L_f^2)B_1\beta^2}{m^2\alpha}\mathbb{E}\left[\sum_{t=1}^T \|\mathbf{u}_t - g(\mathbf{w}_t)\|^2\right]$$

$$+ \frac{396m(C_g^4 L_f^2)}{\alpha B_1}\sum_{t=1}^T \|\mathbf{w}_{t+1} - \mathbf{w}_t\|^2 + \frac{32C_f^2 L_g^2}{\alpha}\mathbb{E}\left[\sum_{t=1}^T \|\mathbf{w}_{t+1} - \mathbf{w}_t\|^2\right]$$

$$+ \frac{2C_g^2 L_f^2}{m}\mathbb{E}\left[\sum_{t=1}^T \|\mathbf{u}_t - g(\mathbf{w}_t)\|^2\right]$$

$$\leq \frac{4}{\alpha}\left\|\mathbf{z}_1 - \frac{1}{m}\sum_{i=1}^m \nabla g_i(\mathbf{w}_1)\nabla f_i(\mathbf{u}_0^i)\right\|^2 + \frac{428mC}{\alpha B_1}\mathbb{E}\left[\sum_{t=1}^T \|\mathbf{w}_{t+1} - \mathbf{w}_t\|^2\right]$$

$$+ \frac{146C_g^2 L_f^2}{m}\mathbb{E}\left[\sum_{t=1}^T \|\mathbf{u}_t - g(\mathbf{w}_t)\|^2\right]$$

We use Lemma 3 to replace $\mathbb{E}\left[\sum_{t=1}^T \|\mathbf{u}_t - g(\mathbf{w}_t)\|^2\right]$:

$$\mathbb{E}\left[\sum_{t=1}^T \|\mathbf{z}_t - \nabla F(\mathbf{w}_t)\|^2\right]$$

$$\leq \frac{4}{\alpha}\left\|\mathbf{z}_1 - \frac{1}{m}\sum_{i=1}^m \nabla g_i(\mathbf{w}_1)\nabla f_i(\mathbf{u}_0^i)\right\|^2 + \frac{428mC}{\alpha B_1}\sum_{t=1}^T \|\mathbf{w}_{t+1} - \mathbf{w}_t\|^2$$

$$+ \frac{146C_g^2 L_f^2 \mathbb{E}\left[\|\mathbf{u}_1 - g(\mathbf{w}_1)\|^2\right]}{B_1\beta} + \frac{1460m^2 C_g^4 L_f^2}{B_1^2\beta}\sum_{t=1}^T \|\mathbf{w}_{t+1} - \mathbf{w}_t\|^2$$

$$\leq \frac{4}{\alpha}\left\|\mathbf{z}_1 - \frac{1}{m}\sum_{i=1}^m \nabla g_i(\mathbf{w}_1)\nabla f_i(\mathbf{u}_0^i)\right\|^2 + \frac{146C_g^2 L_f^2 \mathbb{E}\left[\|\mathbf{u}_1 - g(\mathbf{w}_1)\|^2\right]}{B_1\beta}$$

$$+ \frac{1888m^2 C}{B_1^2\beta}\sum_{t=1}^T \|\mathbf{w}_{t+1} - \mathbf{w}_t\|^2$$

$$\leq \frac{\Delta_0}{\alpha T_0} + \frac{\Delta_0}{\beta T_0} + \frac{1888m^2 C}{B_1^2\beta}\sum_{t=1}^T \|\mathbf{w}_{t+1} - \mathbf{w}_t\|^2$$

Set $\frac{1888m^2 C\eta^2}{B_1^2\beta} \leq \frac{1}{2}$. We have:

$$\mathbb{E}\left[\sum_{t=1}^T \|\mathbf{z}_t - \nabla F(\mathbf{w}_t)\|^2\right] \leq \frac{\Delta_0}{\alpha T_0} + \frac{\Delta_0}{\beta T_0} + \frac{1}{2}\sum_{t=1}^T \|\mathbf{z}_t\|^2$$

According to Lemma 4, we have:

$$\mathbb{E}\left[\sum_{t=1}^T \|\nabla F(\mathbf{w}_t)\|^2\right] \leq \frac{2F(\mathbf{w}_1)}{\eta} + \sum_{t=1}^T \mathbb{E}\left[\|\mathbf{z}_t - \nabla F(\mathbf{w}_t)\|^2\right] - \frac{1}{2}\sum_{t=1}^T \|\mathbf{z}_t\|^2$$

$$\leq \frac{2F(\mathbf{w}_1)}{\eta} + \frac{\Delta_0}{\alpha T_0} + \frac{\Delta_0}{\beta T_0}$$

Finally,

$$\frac{1}{T}\mathbb{E}\left[\sum_{t=1}^T \|\nabla F(\mathbf{w}_t)\|^2\right] \leq \frac{2F(\mathbf{w}_1)}{\eta T} + \frac{\Delta_0}{\alpha T_0 T} + \frac{\Delta_0}{\beta T_0 T}$$

Note that the sample complexity is $\left(B_1 B_2 T + \frac{mnT}{I}\right)$. To ensure the first term and the second term at the same order, we set $I = \left(\frac{mn}{B_1 B_2}\right)$. Also, since we assume that $\alpha I \leq 1$ and $\beta I \leq \frac{m}{B_1}$, we directly set $\alpha = \frac{B_1 B_2}{mn}$ and $\beta = \frac{B_2}{n}$. This setting also satisfies the requirement $B_1 \beta \leq m\alpha$. We also require $\frac{1888 m^2 C \eta^2}{B_1^2 \beta} \leq \frac{1}{2}$. So, we set $\eta = \mathcal{O}(\frac{B_1 \sqrt{B_2}}{m\sqrt{n}})$. With $T = \mathcal{O}\left(\frac{m\sqrt{n}}{B_1 \sqrt{B_2} \epsilon^2}\right)$ and $T_0 = \mathcal{O}\left(\frac{\sqrt{n}}{\sqrt{B_2}}\right)$, We have: $\frac{1}{T}\mathbb{E}\left[\sum_{t=1}^{T} \|\nabla F(\mathbf{w}_t)\|^2\right] \leq \epsilon^2$.

## A.8 Proof of Theorem 5

The analysis is very similar form Theorem 3. We still use Algorithm 3 but employ MSVR-v3 instead. Also, we do not need to decrease $\alpha$, $\beta$, $\eta$ and increase $T$ during each stage. Let's consider the first stage, $4\left\|\mathbf{z}_1 - \frac{1}{m}\sum_{i=1}^{m}\nabla g_i(\mathbf{w}_1)\nabla f_i(\mathbf{u}_0^i)\right\|^2 \leq 4C \leq \mu\epsilon_1$, $\frac{146 C_g^2 L_f^2}{m}\|\mathbf{u}_1 - g(\mathbf{w}_1)\|^2 \leq 146 C \leq \mu\epsilon_1$ and $F(\mathbf{w}_1) - F_* \leq \Delta_F \leq \epsilon_1$, where we set $\epsilon_1 = \max\{\Delta_F, \frac{146 C}{\mu}\}$. Note that in below the numerical subscripts denote the stage index $\{1, \ldots, S\}$. Set $\alpha = \frac{B_1 B_2}{mn}$, $\beta = \frac{B_2}{n}$, $\eta = \mathcal{O}(\frac{B_1 \sqrt{B_2}}{m\sqrt{n}})$ and $T = \mathcal{O}\left(\max\left\{\frac{mn}{B_1 B_2}, \frac{m\sqrt{n}}{\mu B_1 \sqrt{B_2}}\right\}\right)$.

Starting form the second stage, we would prove by induction. Suppose at the stage $s-1$, we have $F(\mathbf{w}_{s-1}) - F_* \leq \epsilon_{s-1}$, $4\left\|\mathbf{z}_{s-1} - \frac{1}{m}\sum_{i=1}^{m}\nabla g_i(\mathbf{w}_1)\nabla f_i(\mathbf{u}_{s-2}^i)\right\|^2 \leq \mu\epsilon_{s-1}$, and $\frac{146 C_g^2 L_f^2}{m}\|\mathbf{u}_{s-1} - g(\mathbf{w}_{s-1})\|^2 \leq \mu\epsilon_{s-1}$. Then at $s$ stage, we have:

$$
\begin{aligned}
F(\mathbf{w}_s) - F_* &\leq \frac{1}{2\mu}\|\nabla F(\mathbf{w}_s)\|^2 \\
&\leq \frac{\epsilon_{s-1}}{\mu\eta T} + \frac{\epsilon_{s-1}}{\alpha T} + \frac{m\epsilon_{s-1}}{\beta B_1 T} \\
&\leq \epsilon_s
\end{aligned}
$$

On the other hand, following the very similar analysis in Theorem 3, we have:

$$
4\left\|\mathbf{z}_s - \frac{1}{m}\sum_{i=1}^{m}\nabla g_i(\mathbf{w}_1)\nabla f_i(\mathbf{u}_{s-1}^i)\right\|^2 \leq \mu\epsilon_s
$$

$$
\frac{146 C_g^2 L_f^2}{m}\|\mathbf{u}_s - g(\mathbf{w}_s)\|^2 \leq \mu\epsilon_s
$$

We proved that $F(\mathbf{w}_s) - F_* \leq \epsilon_s$. That is to say, $F(\mathbf{w}_S) - F_* \leq \epsilon$ when $S = \log_2\left(\frac{2\epsilon_1}{\epsilon}\right) = \log_2\left(\frac{L}{\epsilon}\right)$, and the iteration complexity until this stage is computed as:

$$
\sum_{s=1}^{S} T_s \leq \mathcal{O}\left(\max\left\{\frac{mn}{B_1 B_2}, \frac{m\sqrt{n}}{\mu B_1 \sqrt{B_2}}\right\} \cdot \log\frac{1}{\epsilon}\right)
$$

When $F(\mathbf{w})$ is convex, we define $\hat{F}(\mathbf{w}) = F(\mathbf{w}) + \frac{\mu}{2}\|\mathbf{w}\|^2$. We know that $\hat{F}(\mathbf{w})$ is $\mu$-strongly convex, which implies $\mu$-PL condition. We have proved: for any $\delta > 0$, there exist $T = \mathcal{O}\left(\frac{m\sqrt{n}}{\mu B_1 \sqrt{B_2}} \cdot \log\frac{1}{\epsilon}\right)$ such that $\hat{F}(\mathbf{w}_T) - \hat{F}_* \leq \delta$. It indicates that $F(\mathbf{w}_T) - F_* \leq \delta + \frac{\mu}{2}\|\mathbf{w}_*\|^2 - \frac{\mu}{2}\|\mathbf{w}_T\|^2 \leq \delta + \frac{\mu}{2}D$. For any $\epsilon > 0$, if we choose $\mu = \frac{\epsilon}{D}$ and $\delta = \frac{\epsilon}{2}$, we get $F(\mathbf{w}_T) - F_* \leq \epsilon$, for some $T = \mathcal{O}\left(\frac{m\sqrt{n}}{\epsilon B_1 \sqrt{B_2}} \cdot \log\frac{1}{\epsilon}\right)$.

## B MSVR with Adaptive Learning Rates

Now we show that the proposed MSVR method can be extended to adaptive learning rates and remains the same sample complexity. To use adaptive learning rates, we can revise the weight update step $\mathbf{w}_{t+1} = \mathbf{w}_t - \eta_t \mathbf{z}_t$ in origin MSVR method as follows:

$$
\begin{aligned}
\mathbf{w}_{t+1} &= \mathbf{w}_t - \frac{\eta_t}{\sqrt{\mathbf{h}_t} + \delta}\Pi_{L_f}[\mathbf{z}_t], \\
\mathbf{h}_t' &= (1 - \beta_t')\mathbf{h}_{t-1}' + \beta_t' \mathbf{z}_t'^2,
\end{aligned}
\tag{17}
$$

where $\delta > 0$ is a parameter to avoid dividing zero, $\Pi_{L_f}$ denotes the projection onto the ball with radius $L_f$ and $\mathbf{h}_t = \mathbf{h}_t'$ (Adam-style) or $\mathbf{h}_t = \max\left(\mathbf{h}_{t-1}, \mathbf{h}_t'\right)$ (AMSGrad-style). Inspired by the recent study of Adam-style methods [Guo et al., 2021], we can give the sample complexity of the Adaptive MSVR using similar analysis. We show the proof of adaptive MSVR-v2 for example:

**Theorem 6** *If we choose parameters* $\alpha_{t+1} = \mathcal{O}(\frac{m\eta_t^2}{B_1})$, $\beta_{t+1} = \mathcal{O}\left(\frac{m^2\eta_t^2}{B_1^2}\right)$, $a = O(\frac{mB_2}{B_1})$ *and*

$\eta_t = \mathcal{O}\left(\left(\frac{B_1\sqrt{B_2}}{m}\right)^{2/3}(a+t)^{-1/3}\right)$, *Adaptive MSVR-v2 with learning rate defined in (17), can obtain*

*a stationary point in* $\mathcal{O}\left(\frac{m\epsilon^{-3}}{B_1\sqrt{B_2}}\right)$ *iterations.*

**Remark:** The sample complexity is still at the order of $\mathcal{O}\left(\epsilon^{-3}\right)$. For MSVR-v1 and MSVR-v3, or under the convexity or PL condition, adaptive method can still get the same complexity as the origin rate using a very similar analysis.

**Proof** *Note that since the norm of estimated gradient* $\|\mathbf{z}_t\|$ *is bounded, the value of the learning rate scaling factor* $\mathbf{c} = 1/\left(\sqrt{\mathbf{h}_t} + \delta\right)$ *is also upper bounded and lower bounded, which can be presented as* $c_l \leq \|\mathbf{c}\|_\infty \leq c_u$. *(Note that projection onto a ball of radius* $C_F$ *does not change the analysis, since* $\nabla F$ *is also in this ball.) With this property, We have:*

**Lemma 12** *(Lemma 3 in [Guo et al., 2021]) For* $\mathbf{w}_{t+1} = \mathbf{w}_t - \tilde{\eta}_t\mathbf{z}_t$, *with* $\eta_t c_l \leq \tilde{\eta}_t \leq \eta_t c_u$ *and* $\eta_t L_F \leq c_l/2c_u^2$, *we have following guarantee:*

$$F(\mathbf{w}_{t+1}) \leq F(\mathbf{w}_t) + \frac{\eta_t c_u}{2}\|\nabla F(\mathbf{w}_t) - \mathbf{z}_t\|^2 - \frac{\eta_t c_l}{2}\|\nabla F(\mathbf{w}_t)\|^2 - \frac{\eta_t c_l}{4}\|\mathbf{z}_t\|^2.$$

*Then very similar to the proof to Theorem 2. Denote* $\Gamma_t = F(\mathbf{w}_t) + \frac{B_1}{c_0\eta_{t-1}m}\Delta_t$, *where* $\Delta_t = +\frac{1}{m}\|\mathbf{u}_t - g(\mathbf{w}_t)\|^2 + \left\|\mathbf{z}_t - \frac{1}{m}\sum_{i=1}^m \nabla g_i(\mathbf{w}_t)\nabla f_i(\mathbf{u}_{t-1}^i)\right\|^2$. *We have:*

$$\Gamma_{t+1} - \Gamma_t$$

$$= F(\mathbf{w}_{t+1}) - F(\mathbf{w}_t) + \frac{B_1}{c_0\eta_t m}\Delta_{t+1} - \frac{B_1}{c_0\eta_{t-1}m}\Delta_t$$

$$\leq -\frac{\eta_t c_l}{2}\|\nabla F(\mathbf{w}_t)\|^2 + \frac{\eta_t c_u}{2}\|\mathbf{z}_t - \nabla F(\mathbf{w}_t)\|^2 - \frac{\eta_t c_l}{4}\|\mathbf{z}_t\|^2$$

$$+ \left(\frac{B_1}{c_0\eta_t m} - \frac{B_1^2\beta_{t+1}}{2m^2 c_0\eta_t} - \frac{B_1}{c_0\eta_{t-1}m}\right)\Delta_t + \frac{48C}{c_0\eta_t}\|\mathbf{w}_{t+1} - \mathbf{w}_t\|^2 + \frac{10B_1^2\beta_{t+1}^2 C}{B_2 m^2 c_0\eta_t} + \frac{2\alpha_{t+1}^2 C}{mc_0\eta_t}$$

$$\leq -\frac{\eta_t c_l}{2}\|\nabla F(\mathbf{w}_t)\|^2 - \frac{\eta_t c_l}{4}\|\mathbf{z}_t\|^2 + \frac{64Cc_u}{c_0\eta_t}\|\mathbf{w}_{t+1} - \mathbf{w}_t\|^2 + \frac{14B_1^2\beta_{t+1}^2 Cc_u}{B_2 m^2 c_0\eta_t} + \frac{2\alpha_{t+1}^2 Cc_u}{mc_0\eta_t}$$

$$+ \left(2Cc_u\eta_t + \frac{B_1}{c_0\eta_t m} - \frac{B_1^2\beta_{t+1}}{2m^2 c_0\eta_t} - \frac{B_1}{c_0\eta_{t-1}m}\right)\Delta_t$$

*By setting* $256Cc_u/c_l = c_0$, $\eta_t^2 = \frac{32B_1^2\beta_{t+1}}{m^2 c_0^2 c_l}$, $\alpha_{t+1} = \frac{B_1\beta_{t+1}}{m}$ *and* $B_2 \leq m$, *we have:*

$$\Gamma_{t+1} - \Gamma_t \leq -\frac{\eta_t c_l}{2}\|\nabla F(\mathbf{w}_t)\|^2 + \frac{14B_1^2\beta_{t+1}^2 Cc_u}{B_2 m^2 c_0\eta_t} + \frac{2\alpha_{t+1}^2 Cc_u}{mc_0\eta_t}$$

$$\leq -\frac{\eta_t c_l}{2}\|\nabla F(\mathbf{w}_t)\|^2 + \frac{m^2\eta_t^3 c_0^4 c_l^3}{512 B_2 B_1^2}$$

*This means that, by setting* $\eta_t = \left(\frac{B_1\sqrt{B_2}}{m}\right)^{\frac{2}{3}}(a+t)^{-\frac{1}{3}}$

$$\frac{\eta_T}{2}\mathbb{E}\left[\sum_{t=1}^T\|\nabla F(\mathbf{w}_t)\|^2\right] \leq \frac{\Gamma_1 - \Gamma_{T+1}}{c_l} + \frac{m^2 c_0^4 c_l^2}{512 B_2 B_1^2}\mathbb{E}\left[\sum_{t=1}^T \eta_t^3\right]$$

$$\leq \frac{\Gamma_1}{c_l} + \frac{c_0^4 c_l^2}{16^5}\mathbb{E}\left[\sum_{t=1}^T(a+t)^{-1}\right]$$

$$\leq \frac{\Delta_F}{c_l} + \frac{1}{8\eta_0 c_l} + \frac{c_0^4 c_l^2}{512}\ln(1+T)$$

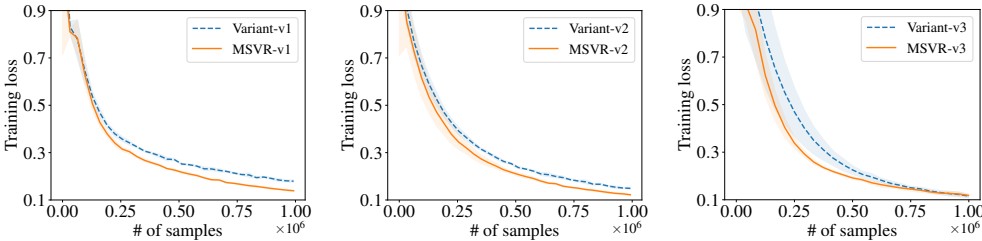

Figure 2: Results for Multi-task AUC Optimization.

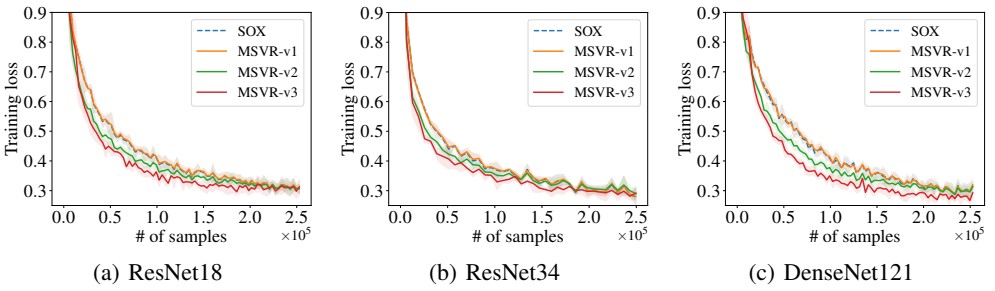

(a) ResNet18        (b) ResNet34        (c) DenseNet121

Figure 3: Results with different networks.

*Denote $M = \frac{\Delta_F}{c_l} + \frac{1}{8\eta_0 c_l} + \frac{c_0^4 c_l^2}{16^5} \ln(1+T)$. Using Cauchy-Schwarz inequality, we have:*

$$\mathbb{E}\left[\sqrt{\sum_{t=1}^{T} \|\nabla F(\mathbf{w}_t)\|^2}\right]^2 \leq \mathbb{E}\left[1/\eta_T\right] \mathbb{E}\left[\eta_T \sum_{t=1}^{T} \|\nabla F(\mathbf{w}_t)\|^2\right]$$

$$\leq \mathbb{E}\left[M\left(\frac{m}{B_1\sqrt{B_2}}\right)^{2/3}(a+T)^{1/3}\right],$$

*Then following the same analysis, we will finally have :*

$$\mathbb{E}\left[\sum_{t=1}^{T} \frac{\|\nabla F(\mathbf{w}_t)\|}{T}\right] \leq \frac{\sqrt{M}(a+T)^{1/6}}{\sqrt{T}}\left(\frac{m}{B_1\sqrt{B_2}}\right)^{1/3} \leq \mathcal{O}\left(\frac{a^{1/6}\sqrt{M}}{\sqrt{T}} + \left(\frac{m}{B_1\sqrt{B_2}T}\right)^{1/3}\right)$$

$$= \mathcal{O}\left(\left(\frac{m}{TB_1\sqrt{B_2}}\right)^{1/3}\right),$$

*where the last inequality is due to $(a+b)^{1/3} \leq a^{1/3} + b^{1/3}$. So, we can achieve the stationary point with $T = \mathcal{O}\left(m/B_1\sqrt{B_2}\epsilon^3\right)$.*

## C    More Experimental Results

In this section, we provide more experimental results and ablation studies. We will consider more applications in the long version of the paper.

### C.1    Ablation Study on Algorithm Design

In this subsection, we conduct the ablation study for our algorithm design. Specially, we verify the effects of our customized error correction term. To compare with traditional variance reduced estimator, we can design an estimator using STORM [Cutkosky and Orabona, 2019] as follows:

$$\mathbf{u}_t^i = \begin{cases} (1-\beta)\mathbf{u}_{t-1}^i + \beta\frac{m}{B_1}g_i\left(\mathbf{w}_t;\xi_t^i\right) + (1-\beta)\frac{m}{B_1}\left(g_i\left(\mathbf{w}_t;\xi_t^i\right) - g_i\left(\mathbf{w}_{t-1};\xi_t^i\right)\right) & i \in \mathcal{B}_1^t \\ (1-\beta)\mathbf{u}_{t-1}^i & i \notin \mathcal{B}_1^t \end{cases} \quad (18)$$

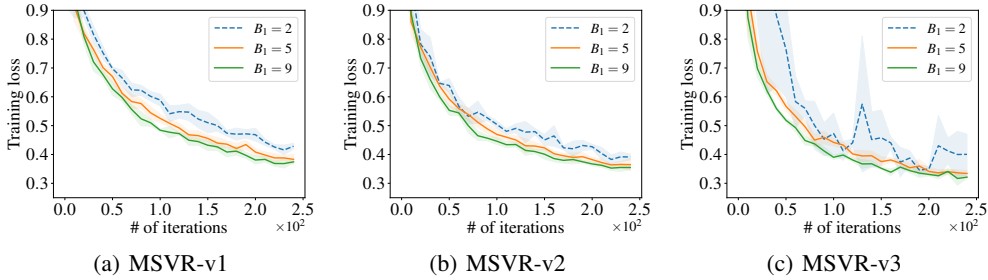

(a) MSVR-v1          (b) MSVR-v2          (c) MSVR-v3

Figure 4: Results with varying $B_1$.

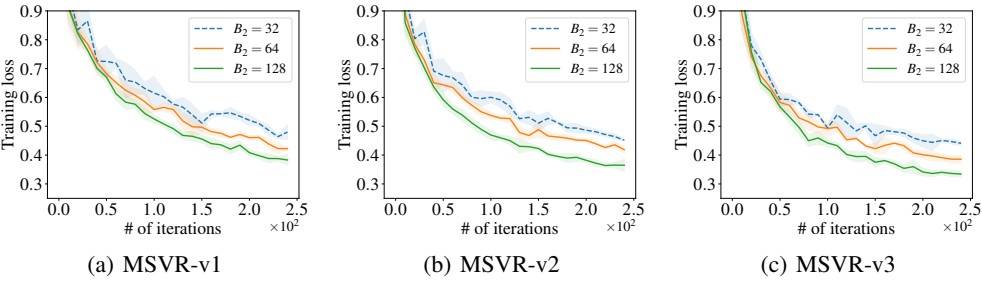

(a) MSVR-v1          (b) MSVR-v2          (c) MSVR-v3

Figure 5: Results with varying $B_2$.

To show the effects of our customized error correction term, we replace the MSVR estimator in our MSVR-v1 and MSVR-v2 algorithm, and use equation (18) instead. We name these two methods Variant-v1 and Variant-v2. For the finite-sum case, we modify the estimator similarly:

$$\mathbf{u}_t^i = \begin{cases} (1-\beta)\mathbf{u}_{t-1}^i + \beta \frac{m}{B_1}\hat{g}_i\left(\mathbf{w}_t; \xi_t^i\right) + (1-\beta)\frac{m}{B_1}\left(g_i\left(\mathbf{w}_t; \xi_t^i\right) - g_i\left(\mathbf{w}_{t-1}; \xi_t^i\right)\right) & i \in \mathcal{B}_1^t \\ (1-\beta)\mathbf{u}_{t-1}^i & i \notin \mathcal{B}_1^t \end{cases} \quad (19)$$

where $\hat{g}_i(\mathbf{w}_t; \xi_t^i) = g_i(\mathbf{w}_t; \xi_t^i) - g_i(\mathbf{w}_\tau; \xi_t^i) + g_i(\mathbf{w}_\tau)$. So, for MSVR-v3, we replace the MSVR estimator with equation (19) and keep other parts unchanged. This new method is named as Variant-v3.

**Results.** We compare different methods on the CIFAR100 dataset and plot the results in Figure 2. As can be seen, all methods perform worse than the origin algorithms, indicating the effectiveness of our customized error correction term in the proposed algorithm.

## C.2   Results with Different Networks

In this subsection, we conduct experiments on SVHN data set with different networks, ResNet18, ResNet34 and DenseNet121, respectively. As can be seen in Figure 3, with all three networks, MSVR-V1 performs closely to SOX, MSVR-v2 converges faster than SOX and MSVR-v1, and the loss of MSVR-v3 decreases most rapidly, indicating the effectiveness of our methods with different networks.

## C.3   Results with Different Batch size

In this subsection, we explore the effect of different batch sizes. First, we fix the inner batch size $B_2 = 128$ and vary $B_1$ in the range $\{2, 5, 9\}$. Then, we fix the outer batch size $B_1 = 5$ and vary $B_2$ in the range $\{32, 64, 128\}$. We conduct the experiments on the Fashion-MNIST data set and show the results in Figure 4 and 5. As can be seen, in terms of iteration complexities, the larger batch size ($B_1$ or $B_2$), the faster the convergence, which is consistent with our theory.