# OpenReview forum: "Multi-block-Single-probe Variance Reduced Estimator for Coupled Compositional Optimization"
_NeurIPS.cc/2022/Conference — NeurIPS 2022 Accept_

### Official Review · Reviewer_qGar · 2022-06-28

**Rating:** 7
**Confidence:** 5
**Soundness:** 4 excellent
**Presentation:** 3 good
**Contribution:** 4 excellent

**Summary:**

This paper considers a general problem: tracking multiple mappings with only O(1) mappings available. An important application investigated in the paper is the Finite-sum Coupled Compositional Optimization (FCCO) problem. By using the proposed Multi-block-Single-probe Variance Reduced (MSVR) estimator, the authors obtain improved complexities for the FCCO problem under different settings. The proposed methods also enjoy better performance in experiments.

**Questions:**

1. Could the analysis also be used in contrastive learning, whose loss function can also be written in the form of FCCO?
2. For the inequality in Lemma1 on page 5 line 174, should it be $\beta_{t+1}$ rather than $\beta_{t}$ ?


**Limitations:**

No problems here.

**Strengths And Weaknesses:**

Overall, the paper is well-written and the contribution is solid.
Strengths:
1. The proposed algorithms have better complexities than the previous best-known method for the FCCO problem, which covers wide applications in ML. The authors investigate the complexities under different settings, i.e., non-convex/convex/strongly-convex/finite-sum. Their theoretical guarantees match the lower bound in terms of $\epsilon$ and $\mu$ in many settings.
2. The authors clearly show the intuition of the proposed method and how to determine the value of the parameter $\gamma_t$. The proposed MSVR estimator is of interest and may inspire the algorithm design for other ML problems.
3. The writing and explanations are clear. The algorithm is simple and the theoretical analysis looks sound to me.

Weakness:
1. The proofs in the supplementary are not very clear-written and more explanations about each lemma are recommended.

---

> ### Author Response · Authors · 2022-08-02
> **Response to Reviewer qGar**
>
> Thank you very much for your constructive comments!
>
> ---
>
> Q1: The proofs in the supplementary are not very clear-written and more explanations about each lemma are recommended.
>
> A1: Thank you for your suggestion. We will give more explanations for each lemma in the supplement.
>
> ---
>
> Q2: Could the analysis also be used in contrastive learning, whose loss function can also be written in the form of FCCO?
>
> A2: Yes, it is possible. One could follow the recent work [Yuan et al., 2022] to extend our analysis to self-supervised contrastive learning.
>
> Reference: Yuan et al. Provable Stochastic Optimization for Global Contrastive Learning: Small Batch Does Not Harm Performance. ICML, 2022.
>
> ---
>
> Q3: For the inequality in Lemma 1 on page 5 line 174, should it be $\beta_{t+1}$ rather than $\beta_t$
>
> A3: Yes, thank you for catching this typo. We have changed the lemma to make it correct (using $\\| \mathbf u_{t}-g(\mathbf w_t) \\|^2$ in the left-hand side) in the revised version.

---

### Official Review · Reviewer_YMmz · 2022-07-08

**Rating:** 7
**Confidence:** 3
**Soundness:** 3 good
**Presentation:** 3 good
**Contribution:** 4 excellent

**Summary:**

In this paper, the authors study the problem of finite-sum coupled compositional optimization (FCCO), in which the inner function may depend on the outer function. When designing stochastic algorithms for FCCO, the main challenge is how to keep track of multiple (inner) functions efficiently. To solve this, the authors propose a new stochastic method named Multi-block-Single-probe Variance Reduced (MSVR) estimator, which could track a sequence of multiple blocks of functions by only probing $O(1)$ blocks. Based on the MSVR estimator, they develop three stochastic algorithms for FCCO, and establish improved complexities for non-convex, convex and strongly convex objectives. Experiments on the multi-task deep AUC maximization show the effectiveness of the proposed methods.

**Questions:**

1. In line 55, the authors claim that “However, this simple change does not improve the complexities over that obtained by Wang and Yang [2022]”. Do you mean that with Eq.4 the complexity is the same complexities as Wang and Yang [2022]?

2. I am confused with the last column of Table 1. Does it denote the ratio between $B_1$ and $B_2$? For BSGD and BSpiderBoost, there are two Big O notations, but for other methods, there is only one Big O notation. What is the exact meaning?

3. After presenting the main theorems, the authors should discuss how to set the values of $B_1$ and $B_2$.


**Limitations:**

Since this paper study an optimization problem, I think that there is no potential negative societal impact of this work.

**Strengths And Weaknesses:**

Strengths:
1.  In the design of MSVR, the authors propose a customized error correction term, which is novel and powerful. Moreover, based on the MSVR, the authors develop three stochastic algorithms and establish optimal sample complexities.
2.  The paper is generally well-written. In particular, the motivation in Section 3.2 is convincing.


Weaknesses:
1. In the experiments, the authors only compare with SOX. I think it is better to add more baselines.
2. Experiments are conducted on only one network (ResNet-18), it is expected to provide more experiments with different models. Moreover,  the parameter analysis and ablation study are required to show the effectiveness of the proposed method.

---

> ### Author Response · Authors · 2022-08-02
> **Response to Reviewer YMmz**
>
> Thank you very much for your constructive comments and suggestions!
>
> We have added more ablation studies in the revision including using different networks and varying $B_1$ and $B_2$  (see Section C in the supplement) and plan to report more experimental results on large-scale data set. We focus on comparing with SOX since it is state of the art for solving the FCCO problem and it has been compared with other baselines (e.g., BSGD) with better performance observed. Below, we address your questions.
>
> ---
>
> Q1: In line 55, the authors claim that “However, this simple change does not improve the complexities over that obtained by Wang and Yang [2022]”. Do you mean that with Eq.4 the complexity is the same complexities as Wang and Yang [2022]?
>
> A1: Yes, we mean by simply using Eq.4, the complexity is on the same order as Wang and Yang [2022], e.g., $\mathcal{O}(m \epsilon^{-4})$ for general smooth case, which is worse than the proposed methods.
>
> ---
>
> Q2: I am confused with the last column of Table 1. Does it denote the ratio between $B_1$ and $B_2$. For BSGD and BSpiderBoost, there are two Big O notations, but for other methods, there is only one Big O notation. What is the exact meaning?
>
> A2: Sorry for using this misleading notation. $B_1$/$B_2$ means $B_1$ and $B_2$. When $B_1$ and $B_2$ are on the same order, e.g. $\mathcal{O}(1)$ for SOX and our method, we only give one Big O notation. We have made them clear in the revised version by using $B_1, B_2$ and stating the order of $B_1$ and $B_2$ explicitly for all methods.
>
> ---
>
> Q3: After presenting the main theorems, the authors should discuss how to set the values of $B_1$ and $B_2$.
>
> A3: Thank you! We have added more discussions in the remark of Theorem 1 in the revision (highlighted in the revision).

---

### Official Review · Reviewer_9gCU · 2022-07-10

**Rating:** 6
**Confidence:** 4
**Soundness:** 3 good
**Presentation:** 3 good
**Contribution:** 2 fair

**Summary:**

The paper considers a scheme in coupled compositional optimization where oracles of inner functions are only available at $\mathcal{O}(1)$ blocks of functions each time. The paper proposes a new estimator called MSVR for inner function values leveraging the idea of variance reduction techniques such as STORM and SVRG. The authors establish the convergence rates of proposed algorithms in the (non)-convex and PL setting. Experimental results on the multi-task deep AUC maximization demonstrate their effectiveness and superiority over SOX - the algorithm proposed previously for solving this problem.


**Questions:**

As stated above I have two **major** questions for this paper:

1. For the Coupled Compositional Optimization (CCO) problem considered, one can treat it as a general compositional optimization problem where a unified outer function $f$ can be defined to operate on $[g_1, g_2, \dots, g_m]$. Moreover, since the outer function considered $f$ is deterministic, one can think of this problem from the perspective of block coordinate updates. It would be much more clear if the authors can elaborate more on their contributions and distinguish them from other existing works on variance reduction and block coordinate updates.

2. The experiments results presented fail to capture the differences between sample complexities of SOX ($\mathcal{O}(\epsilon^{-4})$) and MSVR ($\mathcal{O}(\epsilon^{-2})$) in the finite-sum non-convex setting. The paper should include more experimental results to demonstrate the dependency of $\epsilon, n, B_1, B_2$ instead of simply presenting results when fixing $B_1, B_2$.

In addition, I have a few **minor** comments for this paper:

1. Line 30: Eq. (2) is valid only if each $\xi_{i}$ **uniformly** distributed over a finite support.
2. Line 31: As demonstrated above, I don’t think these problems are different from classical stochastic compositional optimization problems.
3. Table 1: Please indentify $B_1 = |\mathcal{B}_1^t|$
4. Section 2: Please add related works on the variance-reduced block coordinate updates.
5. Line 182: How to ensure that the linearized update in Eq. (6) can obtain an $\mathbf{u}_t^i$ within the range of $g_i$? For example, if $g_i$ is nonnegative given its structure, the linearized update may obtain a negative $\mathbf{u}_t^i$. Would this be a significant issue?
6. Theorem 3: The authors should at least mention the stage-wise design and refer readers to Reference.
7. Line 254: Please cite reference papers discussing the convergence rate under the PL condition.
8. Section 5: It would be more convincing if the authors can further motivate their presented examples. I didn’t see any merit in probing only one task in CIFAR-10 and MNIST datasets.


**Limitations:**

The theoretical and experimental limitations are clearly stated in the previous section.

The paper doesn’t have any potential negative societal impact due to the theoretical nature of the work.


**Strengths And Weaknesses:**

**Strength**: The paper is well-organized and well-written. They leverage the idea of the STORM estimator [1] to obtain better convergence rates compared to [2]. All the results are clearly stated and the proof (although I didn’t check carefully for convex and PL objectives) is sound and consistent with existing literature.

**Weaknesses**: On the theoretical side, the considered problem and proposed algorithms in this paper can be viewed as variance-reduced block coordinate update methods in stochastic compositional optimation. I partially doubt the novelty of this paper though I did not find any related work in this regime. From the practical point of view, the paper needs more experimental results to support the effectiveness and superiority of the proposed algorithm; see comments below.

Reference:

[1] A. Cutkosky and F. Orabona. Momentum-based variance reduction in non-convex SGD. In Advances in Neural Information Processing Systems 32, pages 15210–15219, 2019.

[2] B. Wang and T. Yang. Finite-sum coupled compositional stochastic optimization: Theory and applications. ArXiv e-prints, arXiv:2202.12396, 2022.

---

> ### Author Response · Authors · 2022-08-02
> **Response to Reviewer 9gCU**
>
> Thank you very much for your constructive comments and suggestions! We will revise accordingly.
>
> ---
>
> Q1: It would be much more clear if the authors can elaborate more on their contributions and distinguish them from other existing works on variance reduction and block coordinate updates.
>
> A1: Your vision in terms of block coordinate updates is relevant here. Indeed, Wang and Yang [2022] have explained their tracking of $g=(g_1, \ldots, g_m)$ as stochastic block coordinate updates. In particular, their SOX algorithm views their moving average update, i.e.,
> $$
> \mathbf u_t^i=(1-\beta) \mathbf u_{t-1}^i + \beta g_i(\mathbf w_t; \xi^i_t), i\in\mathcal B_1^t
> $$
>
> as stochastic block coordinate update for the (dynamic) objective $ g_t(\mathbf u)=\sum_{i=1}^m\\|\mathbf{u}^i - g_i(\mathbf w_t)\\|^2/2$. From this perspective, our estimator MSVR can be viewed as applying a momentum-based stochastic block coordinate update for the same objective, with the update
>
> $$
>     q^i_t = \nabla_i g_t(\mathbf u_{t-1};\xi^i_t) + \theta_t (\nabla_i g_t(\mathbf u_{t-1};\xi^i_t) - \nabla_i g_{t-1}(\mathbf u_{t-1};\xi^i_{t})), \quad \mathbf u^i_t = \mathbf u^i_{t-1} - \beta_t q^i_t
> $$
>
> where $\nabla_i g_t(\mathbf u_{t-1};\xi^i_t)=\mathbf u^i_{t-1} - g_i(\mathbf w_t; \xi_t^i)$ and $\theta_t = \gamma_t/\beta_t$. The second term in $q^i_t$ is a momentum term, which is an additional term compared with that of SOX update for $\mathbf u^i_t$.
>
> However,  to the best of our knowledge, there is no prior work analyzing the above momentum-based stochastic block coordinate update. Indeed, our goal is not to optimize $ g_t(\mathbf u)$. Instead we aim to bound $\sum_{t=1}^T\\|\mathbf u_t^i  - g_i(\mathbf w_t)\\|^2$ for a sequence of $\mathbf w_{1}, \ldots, \mathbf w_T$.  Hence, existing methods and analysis on variance reduction and block coordinate updates that focus on optimizing a given fixed objective cannot be applied here.   In another word,  our analysis and its synthesis with the update for FCCO is novel.
>
> ---
>
> Q2: The paper should include more experimental results to demonstrate the dependency of $\epsilon$, $n$, $B_1$, $B_2$ instead of simply presenting results when fixing $B_1$, $B_2$.
>
> A2: We have added more ablation studies by fixing $B_1$ and varying $B_2$, and fixing $B_2$ and varying $B_1$. The results are included in the revision (see Section C in the supplement), which are consistent with our theory, i.e., the larger $B_1$ ($B_2$) the faster the convergence (in terms of iteration complexities). We also plan to report more experimental results on large-scale data set.
>
> ---
>
> Q3: Line 182: How to ensure that the linearized update in Eq. (6) can obtain an $\mathbf{u}_t^i$ within the range of ${g}_i$? For example, if ${g}_i$ is nonnegative given its structure, the linearized update may obtain a negative $\mathbf{u}_t^i$. Would this be a significant issue?
>
> A3: In this paper, we do not restrict the input domain of $f$ or range of $g_i$ for simplicity. If there is a constraint on the range of $g_i$ or input domain of $f$, we can add a projection to project the linearized update into the range of $g_i$, which does not affect our analysis of Lemma 2. Thank you for pointing this out. We have clarified this point in the revision (see remark under Lemma 2).
>
> ---
>
> Q4: Other few minor comments for this paper.
>
> A4: Thank you for your constructive suggestions. We have revised them accordingly.

---

> > ### Comment · Reviewer_9gCU · 2022-08-06
> > **Thank you**
> >
> > Dear authors,
> >
> > Thank you for the response! It clearly addressed all my concerns.

---

### Official Review · Reviewer_v4Z6 · 2022-07-11

**Rating:** 6
**Confidence:** 4
**Soundness:** 3 good
**Presentation:** 3 good
**Contribution:** 3 good

**Summary:**

This paper introduces a novel stochastic method named Multi-block-Single-probe Variance Reduced (MSVR) estimator to track multiple functional mappings across iterations. Based on MSVR, the authors develop several algorithms for solving the finite-sum coupled compositional optimization (FCCO) problem. Theoretical analysis shows that the proposed algorithms obtain optimal complexities across a spectrum of settings.  Experiment results on multi-task deep AUC maximization to demonstrate the advantage of the proposed algorithms.

**Questions:**

1. In Theorem 3, the authors obtain better sample complexities under the $\mu$-PL condition. When considering the $\mu$-PL condition, do we need the convexity condition?

2. There seems an inconsistency in the writing. The authors introduce the concept of PL condition on Page 7, but use strong convexity in other places.

2. The sample complexities in Table 1 increase with $B_2$. It seems that we should use a small value of $B_2$. Is there any benefit of using a large $B_2$?


**Strengths And Weaknesses:**

Strengths:

1. The FCCO problem has a broad spectrum of applications in machine learning and may have a high impact in this area. The proposed algorithms have a major improvement in complexities.

2. The proposed MSVR estimator is novel and quite efficient. It successfully reduces the tracking error with only a constant number of stochastic accesses to functional mappings at each iteration.

3. The sample complexities are much better than existing works, and optimal in many cases. The experimental results are also convincing.

Weaknesses:
1. Besides multi-task deep AUC maximization, more experiments on others problems could be added to evaluate the proposed algorithms.

2. The MSVR estimator seems incremental with respect to STORM, although the parameter is set in a different way.

---

> ### Author Response · Authors · 2022-08-02
> **Response to Reviewer v4Z6**
>
> Thank you for the review!
> We would like to clarify that although the MSVR estimator is incremental to STORM on the surface, our analysis is novel and the improvements for the FCCO problem are significant compared with state of the art results. Since we focus more on theoretical analysis covering non-convex, convex, strongly convex and PL objectives, our experiments have been focused on one application with six benchmark datasets. We have added more ablation studies in the revision (See Section C in the supplement).  We will consider more applications in the long version of the paper. Below, we address your questions.
>
> ---
>
> Q1: When considering the $\mu$-PL condition, do we need the convexity condition?
>
> A1: No, we do not need the convexity condition when considering the $\mu$-PL condition.
>
> ---
>
> Q2: The authors introduce the concept of PL condition on Page 7, but use strong convexity in other places.
>
> A2: Since the PL condition is weaker than strong convexity, so the results for the PL condition are directly applicable to strong convexity (Note that if a function is $\mu$-strong convex, it would also satisfy the $\mu$-PL condition). We have mentioned the results under the PL condition in the abstract and introduction part in the revision.
>
> ---
>
> Q3: It seems that we should use a small value of $B_2$. Is there any benefit of using a large $B_2$?
>
> A3:  It is true that  smaller $B_2$ is better for sample complexities. However, There is **benefit** of using large $B_2$ in terms of iteration complexity. The larger $B_2$, the smaller the iteration complexity. Please check Theorem 1 and Theorem 2 for the iteration complexities.  Hence, from the computational perspective, if $B_2$ samples can be processed in parallel (e.g., in GPU), there is a benefit of using large $B_2$. In our experiments, we use $B_2=128$.

---

> > ### Comment · Reviewer_v4Z6 · 2022-08-07
> > **Thank you for your detailed explanation.**
> >
> > Dear Authors,
> >
> > Thanks for your detailed comments. It makes more sense now.

---

### Meta-Review · Area_Chair_joHH · 2022-08-22

**Recommendation:** Accept
**Confidence:** Certain

**Metareview:**

The paper makes a nice contribution to the growing field of stochastic compositional optimization. In particular, it considers the case of coupled compositional problems and provides an algorithm that tracks all the inner-level objective information required in an efficient manner. Sample complexities (which are intuitively, optimal) are established.

The authors **must**  emphasize that they work under the stronger Assumption 3 in the revision.

**Award:**

No

---

### Decision · Program_Chairs · 2022-09-14

Accept